# Understanding the Role of Invariance in Transfer Learning

**Till Speicher**                                                    *tspeicher@mpi-sws.org*
*MPI-SWS*

**Vedant Nanda**                                                     *vnanda@mpi-sws.org*
*MPI-SWS*

**Krishna P. Gummadi**                                               *gummadi@mpi-sws.org*
*MPI-SWS*

**Reviewed on OpenReview:** *https://openreview.net/forum?id=spJI4LSPIU*

## Abstract

Transfer learning is a powerful technique for knowledge-sharing between different tasks. Recent work has found that the representations of models with certain invariances, such as to adversarial input perturbations, achieve higher performance on downstream tasks. These findings suggest that invariance may be an important property in the context of transfer learning. However, the relationship of invariance with transfer performance is not fully understood yet and a number of questions remain. For instance, how important is invariance compared to other factors of the pretraining task? How transferable is learned invariance? In this work, we systematically investigate the importance of representational invariance for transfer learning, as well as how it interacts with other parameters during pretraining. To do so, we introduce a family of synthetic datasets that allow us to precisely control factors of variation both in training and test data. Using these datasets, we a) show that for learning representations with high transfer performance, invariance to the right transformations is as, or often more, important than most other factors such as the number of training samples, the model architecture and the identity of the pretraining classes, b) show conditions under which invariance can harm the ability to transfer representations and c) explore how transferable invariance is between tasks. The code is available at `https://github.com/tillspeicher/representation-invariance-transfer`.

## 1 Introduction

Many learning problems are increasingly solved by adapting pretrained (foundation) models to downstream tasks (Bommasani et al., 2021). To decide when and how pretrained models can be transferred to new tasks, it is important to understand the factors that determine the transfer performance of their representations. The literature on transfer learning has proposed a number of factors that influence transfer performance. Among them are for instance the accuracy of a model on its training dataset, the architecture and size of the model and the size of the training dataset (Kolesnikov et al., 2020; Huh et al., 2016; Kornblith et al., 2019). One surprising recent result is that models robust to adversarial attacks, *i.e.* invariant to certain $\epsilon$-ball perturbations, exhibit higher downstream performance on transfer tasks than non-robust ones, despite achieving lower performance on their training dataset (Salman et al., 2020). Other invariances, such as to textures, have also been found to boost performance (Geirhos et al., 2018a). These findings suggest *invariance* as another important factor that can influence transfer performance.

A model can be said to be invariant to some transformation if its output or representations do not change in response to applying the transformation to its input. Invariance has been recognized as an important property of models and their representations and consequently has received a lot of attention (Cohen et al., 2019; Bloem-Reddy & Teh, 2020; Lyle et al., 2020; Ericsson et al., 2021b). Most of this work, however,

focuses on the role that invariance plays for a specific task. In the case of transfer learning, on the other hand, there are two tasks involved: the task on which the model is trained and the task to which it is transferred. The effect of invariance, both learned during pretraining and required by the downstream task, on the *transfer performance* of representations has received less attention and is not fully understood yet.

One reason why the relationship between invariance and transfer performance has not been more thoroughly explored is that doing so is challenging, especially when only using common real-world datasets. To investigate invariance at a fine-grained level, it is necessary to know the different ways in which inputs to a model differ, in order to determine how those differences relate to changes in representations. This, however, is not possible with typical real-world datasets such as CIFAR-10 (Krizhevsky et al., 2009), ImageNet (Russakovsky et al., 2015) or VTAB (Zhai et al., 2020). For example, the CIFAR-10 dataset contains images of cats, but using this dataset to assess whether a model is invariant to the position or pose of cats is not possible, since no position or other information is available beyond class labels.

Therefore, to study invariance carefully, we introduce a family of synthetic datasets, Transforms-2D, that allows us to precisely control the differences and similarities between inputs in a model's training and test sets. Using these datasets, we explore the importance of invariance in achieving high transfer performance, as well as how transferable invariance is to new tasks. Concretely, we make the following contributions:

- We introduce a family of synthetic datasets called Transforms-2D, which allows us to carefully control the transformations acting on inputs. By using these datasets, we are able to train models to exhibit specific invariances in their representations and to evaluate their performance on transfer tasks that require specific invariances. We also use them as the basis for measuring invariance to input transformations.

- We investigate the connection between invariance and downstream performance and compare it to other factors commonly studied in the transfer learning literature, such as the number of training samples, the model architecture, the relationship of training and target classes, and the relationship of training- and target-task performance. We find that while these other factors play a role in determining transfer performance, sharing the invariances of the target task is often as or more important. We further show how undesirable invariance can harm the transfer performance of representations.

- We explore the transferability of invariance between tasks and find that in most cases, models can transfer a high degree of learned invariance to out of distribution tasks, which might help explain the importance of invariance for transfer performance.

- While our observations are derived from experiments on synthetic data, we validate them on real-world datasets and find that similar trends hold in these settings.

## 2 Related Work

**Transfer learning** is a well studied topic in the machine learning community. Prior work has identified a number of factors that contribute to transfer performance, such as model size (Kolesnikov et al., 2020; Abnar et al., 2021), size and characteristics of the pretraining dataset (Huh et al., 2016; Kornblith et al., 2019; Azizpour et al., 2015; Neyshabur et al., 2020; Entezari et al., 2023) and adversarial robustness (Salman et al., 2020). In this work, we investigate the impact of invariance on transfer performance more broadly as another dimension and compare its effect to the aforementioned factors. In this context, prior work has found that DNNs can have difficulties generalizing certain invariances (Azulay & Weiss, 2018; Zhou et al., 2022). Our work also aims to understand this phenomenon better by investigating conditions under which invariance transfers more closely.

**Invariance** and equivariance have been studied in the context of representation learning with the goals of better understanding their properties (Kondor & Trivedi, 2018; Bloem-Reddy & Teh, 2020; Lyle et al., 2020; Cohen et al., 2019; Von Kügelgen et al., 2021), measuring them (Goodfellow et al., 2009; Fawzi & Frossard, 2015; Gopinath et al., 2019; Kvinge et al., 2022; Nanda et al., 2022), leveraging them for contrastive learning (Wang & Isola, 2020; Ericsson et al., 2021a) and building in- and equivariant models (Benton et al.,

2020; Cohen & Welling, 2016; Zhang, 2019; Weiler & Cesa, 2019). Our work is also attempting to understand the implications and benefits of invariant models better. However, most prior work on understanding invariance analyzes how invariance benefits a particular task or is focussed on a specific domain (Ai et al., 2023), whereas we are interested in understanding the relationship of invariance between different tasks more broadly. Additionally, we complement the theoretical perspective that many prior works are offering with an empirical analysis.

**Data augmentations** have been studied as a tool to improve model performance (Perez & Wang, 2017; Shorten & Khoshgoftaar, 2019; Cubuk et al., 2018), and to imbue models with specific invariances (Ratner et al., 2017). Their effects have also been thoroughly investigated (Perez & Wang, 2017; Chen et al., 2020a; Huang et al., 2022; Geiping et al., 2022; Balestriero et al., 2022). In our work we leverage data augmentations to both train models with specific invariances as well as to evaluate the degree and effect of invariance in their representations.

In **self-supervised learning (SSL)**, models are trained to be invariant to certain data augmentations in order to obtain pseudo labels (Doersch et al., 2015; Zhang et al., 2016) or to introduce contrast between similar and dissimilar inputs (Chen et al., 2020b; Grill et al., 2020; Kim et al., 2020; Ericsson et al., 2021b). However, invariance is often treated in an ad hoc and opportunistic manner, *i.e.* data transformations are selected based on how much they boost the validation performance of models. Our findings complement work that uses invariance as a building block, by assessing the importance of invariance to data transformations, relative to other factors such as the model architecture.

The **robustness** literature has also studied invariance extensively in order to safeguard model performance against adversarial perturbations (Szegedy et al., 2013; Papernot et al., 2016), natural image corruptions (Geirhos et al., 2018b; Hendrycks & Dietterich, 2019; Taori et al., 2020) or distribution shift (Recht et al., 2019). This line of work is similar in spirit to ours, as it also shows that invariance, or a lack thereof, can have a significant impact on model performance. However, robustness research is primarily interested in avoiding performance degradations when specific transformations are introduced to a dataset, rather than understanding how the ability to transfer representations between datasets depends on their invariance to transformations. Some prior works have found that too much invariance can be detrimental to the robustness of models (Jacobsen et al., 2018; Kamath et al., 2019; Singla et al., 2021). We also investigate this phenomenon and expose conditions under which representational invariance can harm transfer performance. Additionally, the field of invariant risk minimization has investigated robustness to changes in spurious correlations between different domains (Muandet et al., 2013; Arjovsky et al., 2019).

**Synthetic datasets.** There have been proposals for fully synthetic datasets that allow for a more careful study of the properties of representations (Hermann & Lampinen, 2020; Matthey et al., 2017). Our work leverages previous work by Djolonga et al. (2021) and uses it to construct a dataset that allows for precise control over variations in the data.

## 3 Controlling and Evaluating Invariance in Representations

We begin by describing our approach to controlling for and evaluating the presence of invariance in representations.

### 3.1 Terminology

**Notation.** We operate in the standard supervised learning setting and denote by $\mathcal{D} = \{(\boldsymbol{x}_i, y_i)_{i=1}^N\}$ a dataset consisting of $N$ examples $\boldsymbol{x} \in \mathcal{X} \subseteq \mathbb{R}^n$ and associated labels $y \in \mathcal{Y} = \{1, \ldots, K\}$. The task is to find a function $g : \mathcal{X} \mapsto \mathcal{Y}$ that minimizes the empirical risk on $\mathcal{D}$. $g$ is a neural network that is trained by minimizing the categorical cross-entropy loss over its predictions. For our purpose it is convenient to write $g$ as $g = g_{cls}(g_{rep}(.))$ where $g_{rep} : \mathcal{X} \mapsto \mathcal{Z}$ maps inputs $\boldsymbol{x}$ to representations $\boldsymbol{z} \in \mathcal{Z} \subseteq \mathbb{R}^m$ and $g_{cls} : \mathcal{Z} \mapsto \mathcal{Y}$ maps representations to predictions $\hat{y}$. We will refer to $g_{rep}$ simply as $g$ if the meaning is clear from the context.

We primarily focus on representations at the penulatimate layer that are fixed after pretraining in this work. We study fixed representations, since retraining $g_{rep}$ would change the invariance properties of the representations, and thus would not allow us to cleanly answer the question of how the invariance properties of representations affect their downstream performance. We focus on the penulatimate layer since its representations are most commonly used for transfer learning under the linear probing regime (Kornblith et al., 2019; Salman et al., 2020; Chen et al., 2020b).

**Invariance.** We are interested in the invariance of representations to transformations in a model's input. In the context of this work, we define invariance as follows: Given a transformation $t : \mathcal{X} \mapsto \mathcal{X}$, we say that model $g$ is *invariant* to $t$ if $g(t(x)) = g(x)$ for all $x \in \mathcal{X}$. We say that $g$ is invariant to a set of transformations $T$ if it is invariant to all transformations $t \in T$.

### 3.2 Constructing Invariant Representations

The main approaches to construct invariant representations are training with data augmentations (Cubuk et al., 2018; Antoniou et al., 2017; Benton et al., 2020; Chen et al., 2020a) and architectures that are equi- or invariant by construction (Cohen & Welling, 2016; Zhang, 2019). The data augmentation approach randomly applies certain transformations to the input data during training. Since the transformations are independent from the training data, models trained this way have to become invariant to them in order to minimize their loss. Invariant architectures on the other hand build specific inductive biases into the model architecture, that make them equi- or invariant to certain transformations, such as rotation or translation (Worrall et al., 2017).

In this work, we choose data augmentations — *i.e.* input transformations — as the method to construct invariant representations. Our choice is motivated by the fact that a) training with data transformations is the most commonly used method to construct invariant representations in the literature, b) it is flexible and allows us to construct invariances to any type of transformation that we can define through code (as opposed to architectural invariance, which requires a different model design for each type of invariance) and c) it allows us to leverage the same mechanism we use to train invariant networks to also evaluate the invariance of representations. In Section 5 we show that training with input transformations indeed leads to representations that are invariant to those transformations.

### 3.3 Controlling Data Transformations via Synthetic Data

To both construct representations with known invariance properties and to evaluate them on tasks with known invariance requirements, we need to be able to control the transformations present in a model's training and test data. For instance, to determine whether the representations of a model are invariant to a particular transformation, it is necessary to probe it with inputs that only differ by this transformation.

Using real-world datasets for this task is very challenging for two reasons. First, information about how inputs are transformed relative to each other, for example whether objects in images differ by certain translations or rotations, is typically not available in most real-world datasets, beyond coarse-grained information like the class that an input belongs to. Second, even if such annotations were available for real-world data, they would come with the risk of confounders between transformations and other data factors, such as the objects present in the data. For example, images of huskies might all have been taken on snowy background (Ribeiro et al., 2016). To carefully control for such confounders, data would have to be sampled in a randomized manner in a lab setting, thus diminishing realism benefits.

Therefore, in order to properly study invariance in representations, we introduce a family of synthetic image datasets that allows us to precisely control which objects are present in images, as well as the transformations acting on them. We call it the family of *Transforms-2D* datasets.

A Transforms-2D dataset $\mathcal{D}(O, T)$ is defined by a set of foreground objects $O$ and a set of transformations $T$, with associated distributions $P_O$ and $P_T$ over $O$ and $T$, respectively. In addition, there is a set $B$ of background images with uniform distribution $P_B$, which is the same across all datasets in the family. To sample an image from $\mathcal{D}(O, T)$, we sample an object $o \sim P_O$, a transformation $t \sim P_T$, and a background $b \sim B$, and then create an image as $b(t(o))$, where $t(o)$ is the transformed object and $b(.)$ denotes pasting

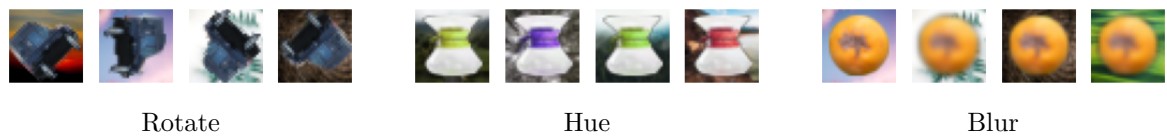

| Rotate | Hue | Blur |

Figure 1: [**Transforms-2D examples.**] Example images sampled from the Transforms-2D dataset. Each row shows a different transformation being applied to one of the object prototypes.

onto the background $b$. Each object $o \in O$ defines a class in the dataset, with each sample having $o$ as its class. That means that each class is based on a single object prototype $o \in O$, and different instances of the same class are created by applying transformations from $T$ to the prototype. Sample images for different transformations are shown in Figure 1.

**Foreground objects and background images.** Each sample from a $\mathcal{D}(O, T)$ dataset consists of a transformed foreground object pasted onto a background image. Foreground objects $O$ are a subset of 61 photographs of real-world objects, such as food and household items, vehicles, animals, etc. Each object image has a transparency masks, such that it only contains the object and no background. $P_O$ samples objects uniformly at random from $O$. We use the same set $B$ of background images for each $\mathcal{D}(O, T)$, which are photographs of nature scenes, chosen uniformly at random from a set of 867 candidates. The foreground and background images are based on the SI-score dataset (Djolonga et al., 2021). We subsample images to have a resolution of 32 x 32 pixels, since this size provides enough detail for models to be able to distinguish between different objects, even with transformations applied to them, while allowing for faster iteration times in terms of training and evaluation.

**Transformations.** A transformation $t$ is typically a combination of multiple transformations of different types, *e.g.* a translation followed by a rotation. Therefore, the set of transformations $T$ is the Cartesian product of sets of different transformation types, *i.e.* $T = T^{(1)} \times \ldots \times T^{(k)}$, and transformations $t \in T$, $t = t^{(1)} \circ \ldots \circ t^{(k)}$ are concatenations of the transformations of each type $t^{(i)} \in T^{(i)}$. We denote the cardinality of $T$ by the number of transformation types that it uses, *i.e.* if $T = T^{(1)}, \ldots, T^{(k)}$, then $|T| = k$. To sample a transformation $t \sim P_T$, we sample transformations $t^{(i)} \sim P_{T^{(i)}}$ for each type $i$ and concatenate them to form $t$. We use three categories of transformation types that span a comprehensive set of image manipulations: i) *geometric* (translate, rotate, scale, vertical flip, horizontal flip, shear), ii) *photometric* (hue, brightness, grayscale, posterize, invert, sharpen), and iii) *corruption* (blur, noise, pixelate, elastic, erasing, contrast). Additional information on and examples of the transformations, including information about their distributions $P_{T^{(i)}}$ can be found in Appendix A.1.

In our experiments we use 50,000 training samples to train models with specific invariances, as well as 10,000 validation and 10,000 test samples, if not stated differently. These numbers mimic the size of the CIFAR datasets (Krizhevsky et al., 2009).

### 3.4 Measuring Invariance in Representations

In order to measure how invariant representations are to specific transformations $T$, we measure how much they change in response to applying transformations from $T$ to the input, *i.e.* how *sensitive* representations are to transformations from $T$. Given a model $g$, a set of transformations $T$ and objects $O$, we measure the sensitivity of $g$ to $T$ based on the L2-distance as

$$sens(g|T, O) = \frac{1}{C} \mathop{\mathbb{E}}_{x_1, x_2 \sim \mathcal{D}(T, O)} \big[\, \|g(x_1) - g(x_2)\|_2 \,\big] \tag{1}$$

where $x_1, x_2 \sim \mathcal{D}(T, O)$ are pairs sampled from $\mathcal{D}(T, O)$ as $x_1 = b(t_1(o))$ and $x_2 = b(t_2(o))$, for $o \sim P_O$, $t_1, t_2 \sim P_T$ and $b \sim P_B$, *i.e.* $x_1$ and $x_2$ only differ in the transformation applied to them. $C$ is a normalization constant, which measures the average distance between any two random samples from $\mathcal{D}(T, O)$, *i.e.* $C = \mathop{\mathbb{E}}_{x, x' \sim \mathcal{D}(T, O)} \big[\, \|g(x) - g(x')\|_2 \,\big]$, without any sampling constraints on $x, x'$. Intuitively, $sens(g|T, O)$ measures the ratio of the distance between two samples that only differ in their transformation, relative to

the average distance between any two samples from $\mathcal{D}(T, O)$. The lower $sens(g|T, O)$, the more invariant the representations are to the transformations in $T$. In our experiments we approximate each of the expectations as an average over 10,000 sample pairs.

## 4 How Important is Representational Invariance for Transfer Learning?

We want to understand the factors that determine whether a representation trained on one dataset transfers, *i.e.* performs well, on another dataset. In particular, we are interested in understanding how important invariance is for transfer performance, compared to other factors, and whether the wrong invariances can harm transfer performance.

### 4.1 How Important is Invariance Compared to Other Factors?

To better understand how important representational invariance is compared to other factors, we leverage the Transforms-2D dataset to create training and test tasks with known required invariances. In particular, we compare invariance against the following factors: *dataset size*, *model architecture*, and the *number and identity of classes*.

We set up experiments that mimic the typical transfer learning setting. In each experiment, we sample disjoint sets of training and evaluation objects $O_t$ and $O_e$ for the source and target task, respectively, as well as disjoint sets of training and evaluation transformations $T_t$ and $T_e$. Using these sets we create datasets as described below and train models on a training task, freeze their weights and transfer their penultimate layer representations to a target task, where we train a new linear output layer. Both the training and target task are classification problems based on the Transforms-2D dataset, which differ in the set of objects that need to be classified. For our experiments, we set $|O_t| = |O_e| = 30$ (roughly half the set of 61 available objects) and $|T_t| = |T_e| = 3$, with transformation types sampled uniformly among the 18 available types of transformations in Transforms-2D.

**Effect of invariance:** To measure the effect of invariance on transfer performance, we train pairs of models on two versions of the training dataset, $\mathcal{D}_s = \mathcal{D}(O_t, T_e)$ and $\mathcal{D}_d = \mathcal{D}(O_t, T_t)$, respectively, whose only difference is that $\mathcal{D}_s$ uses the *same* transformations $T_e$ as the target dataset, while $\mathcal{D}_d$ uses the *disjoint* set of training transformations $T_t$. This setup provides us with a "same-transformations" and a "different-transformations" model $g_s$ and $g_d$, respectively, which only differ in the transformations in their training dataset and thus the invariances in their representations. Comparing the performance of these two models on the target dataset $\mathcal{D}_e = \mathcal{D}(O_e, T_e)$ after fine-tuning allows us to quantify how important having the right invariances is for the target task.

For example, the target task might transform objects by rotating, blurring and posterizing them ($T_e$). In order to perform well on this task ($\mathcal{D}_e$), a model needs to learn representations that are invariant to these transformations. Models $g_s$ that are pretrained on source datasets $\mathcal{D}_s$ with the same transformations $T_e$ acquire these invariances, whereas models $g_d$ that are trained on datasets $\mathcal{D}_d$ with disjoint transformations $T_t$, *e.g.* on a dataset where objects are translated, scaled and color inverted, would not acquire the invariances required for the target task.

**Effect of other factors:** To compare the effect of invariance to that of the other factors, (*e.g.* the number of training samples) we train multiple such pairs of models $g_s$ and $g_d$. Each pair is trained on datasets $D_s$ and $D_d$ that both use a different value of the factor that we want to compare to. Comparing the within-pair with the between-pair performance differences allows us to quantify how important invariance is compared to these other factors. Details on the training and evaluation procedure can be found in Appendix B.

- **Dataset size**: The size of the training dataset is typically considered an important factor in determining how useful representations are for downstream tasks. We compare its effect to invariance by training each pair of models with a different number $n$ of training samples that are drawn from $\mathcal{D}_s$ and $\mathcal{D}_d$. We use $n \in \{1000, 10000, 50000, 100000, 500000\}$.

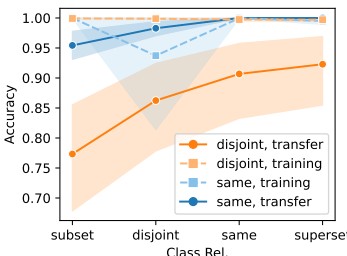 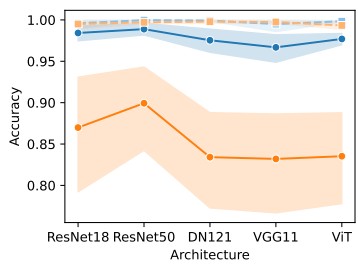 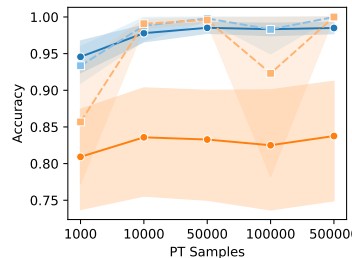

(a) Invariance vs class relationship.    (b) Invariance vs architecture.    (c) Invariance vs number of training samples.

Figure 2: [**Impact of invariance vs other factors on transfer performance in Transforms-2D.**] Training (dotted lines) and transfer performance (solid lines) for models trained with different factors of variation and different invariances on the Transforms-2D dataset. Models trained to be invariant to the same transformations as the target tasks (blue) transfer significantly better than models trained to be invariant to different transformations (orange). This effect is very strong compared to the effect of other factors, such as the number of training samples, the model architecture or the relationship between the training and target classes. The reported numbers are aggregated over 10 runs.

- **Model architecture**: The architecture and capacity of models is important for their performance, with larger models typically performing better. To compare the effect of architecture and model size, we train multiple pairs of models $g_s, g_d$, each with a different architecture. Here, we use ResNet-18, ResNet-50, Densenet-121, VGG-11 and Vision Transformer (ViT). For more details, see Appendix B.1.

- **Number and identity of classes**: In transfer learning, the objects in the training and target domain typically differ. To understand how much impact the difference of training and target objects has compared to invariance, we construct four pairs of training datasets whose objects $O_t$ are related in different ways to the target objects $O_e$: a subset $O_{sub} \subset O_t$, $|O_{sub}| = \frac{1}{3}|O_t| = 10$, a disjoint set with the same cardinality $O_{disj} \cap O_t = \emptyset$ and $|O_{disj}| = |O_t| = 30$, the same $O_{same} = O_t$ and a superset $O_{sup} \supset O_t$ of $O_t$ with $|O_{sup}| = 2 * |O_t| = 60$.

**Validation on real-world data:** The experiments on Transforms-2D data allow us to cleanly disentangle the effect of invariance from the other factors, but come with the risk of being limited to synthetic data. To test whether our observations hold up under conditions closer to real-world settings, we perform similar experiments as those on Transforms-2D on the CIFAR-10 and CIFAR-100 datasets. We cannot control transformations directly in these datasets, but we can approximate the setup described above by applying the transformations from Transforms-2D as data augmentations. As before, we pretrain models on two versions of the CIFAR datasets, one with the same transformations/data augmentations as the target task and the other with a disjoint set of transformations. We use a subset of 5 classes for CIFAR-10 and 50 classes for CIFAR-100 for pretraining and study transfer performance on the other half of the classes. Again, we compare the effect of using the same vs a disjoint set of invariances as the target task to the effect of varying the number of training samples, the model architecture and the class relationship.

**Results:** Figure 2 compares the effect of invariance on transfer accuracy with that of the other factors (number of training samples, architecture, class relationship), on the Transforms-2D dataset. The accuracy of all models on their training tasks is very similar. However, when transferred to the target task, the models that were trained with the same transformations as the target task (*i.e.* to have the invariances required by the target task) outperform the models that were trained with a disjoint set of transformations by a significant margin in all cases. We show analogous results for the CIFAR-10 and CIFAR-100 datasets in Appendix C.1 in Figures 7 and 8.

Figure 3 quantifies the difference in transfer performance caused by invariance and compares it to the differences caused by the other factors for the Transforms-2D, as well as the CIFAR-10 and CIFAR-100

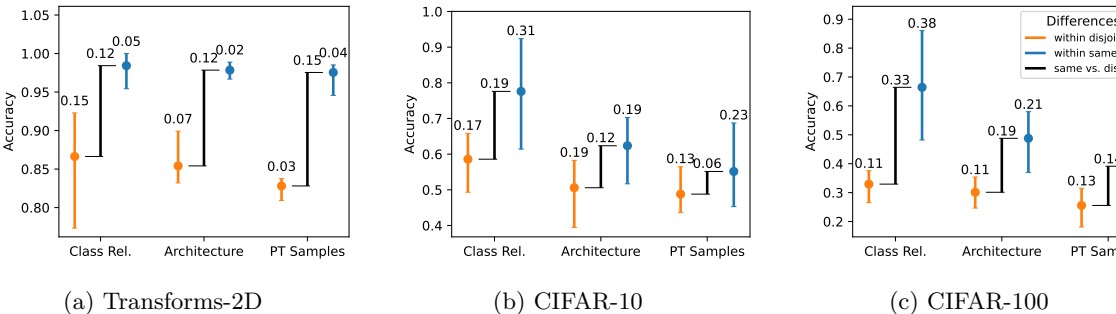

| (a) Transforms-2D | (b) CIFAR-10 | (c) CIFAR-100 |

Figure 3: [**Difference in transfer performance due to invariance vs other factors.**] We compare the differences in transfer performance caused by representational invariance with the differences caused by changes to other factors on the Transforms-2D and the CIFAR-10 and CIFAR-100 datasets (with data augmentations). Orange (blue) bars show the span of transfer performance for different-transformation models $g_d$ (same-transformation models $g_s$), for each comparison factor (class relationship, architecture, number of samples). Black bars show the difference between transfer performance means across factor values for $g_d$ and $g_s$, *i.e.* the difference in performance caused by having the same vs different invariances as the target task. Across all datasets, the difference in transfer performance due to representational invariance is comparable and often larger than the difference due to varying the other factors.

datasets. For Transforms-2D, the difference attributable to invariance is larger than that attributable to all other factors, with the exception of the different-transformation model $g_d$ for class-relationship, where it is comparable. For CIFAR-10 and CIFAR-100, the difference in transfer performance due to invariance is comparable to the difference due to class relationship and architecture. It is similar or lower than the difference due to the number of training samples, but still substantial.

In the latter case of sample counts, it is worth noting that in the CIFAR datasets, classes consist of a diverse set of images that can be seen as different transformations of the same object (*e.g.* cars or birds that differ in their model or species, color, camera angle, background, etc.). With more samples per class, the models likely see a more diverse set of transformations of the same object and thus might be able to become more invariant to irrelevant differences in object appearance. Therefore, comparing invariance due to data augmentations with the number of training samples, might implicitly compare explicit and implicit representational invariance. This relationship highlights the difficulties in studying the effect of invariance in uncontrolled real-world settings.

In Appendix C.2 we investigate how full-model fine-tuning affects the invariance of representations, and whether fine-tuning the whole model can change the invariances learned during pretraining. We find that with a small amount of fine-tuning data, representations that were pretrained with the right invariances still outperform ones that were pretrained with different invariances. The more data models are fine-tuned on, *i.e.* the more fine-tuning approximates re-training the model on the target dataset, the more the the advantage of pretraining with the right invariances diminishes.

Taken together, when transferring representations between tasks, our results show that invariance is as important or more important than other commonly studied factors in most cases. Our findings have important implications for model pretraining and pretraining dataset selection, domains that are especially relevant in an era where the usage of pretrained foundation models is becoming very prevalent (Bommasani et al., 2021). Practitioners should pay special attention to the variations present in pretraining data, as well as to data augmentations, since both can strongly influence representational invariance and thus downstream performance. For instance, the high importance of invariance compared to the number and type of classes in the pretraining data shown in Figure 2a and Figure 3 means that when creating pretraining datasets, it may be beneficial to allocate more effort towards obtaining a larger diversity of object transformations compared to sampling more objects.

## 4.2 Can Invariance be Exploited to Harm Transfer Performance?

The previous results highlight that having the right representational invariances can significantly benefit transfer performance. On the flip-side, they also show that the wrong invariances harm transfer performance. Prior work has demonstrated similar detrimental effects for certain types of excessive invariance (Jacobsen et al., 2018). An interesting question therefore is: Could an adversary exploit invariance to harm transfer performance on specific downstream tasks?

To answer this question, we combine our Transforms-2D data with the CIFAR-10 dataset (Krizhevsky et al., 2009), such that either the information in the Transforms-2D data or in CIFAR-10 is irrelevant for the target task. Concretely, we augment the CIFAR-10 dataset by pasting small versions of the Transforms-2D objects described in section 3.3 onto the images in a completely random manner, *i.e.* uncorrelated with the CIFAR-10 labels.

**Notation.** We denote by $X$ the features available in the input and by $Y$ the category of labels that the model is trained to predict. $C$ stands for CIFAR-10 and $O$ for objects from Transforms-2D. $X = C$ means that the input data is CIFAR backgrounds, and $Y = C$ means that the task is to classify images based on the CIFAR classes. $X = C + O, Y = C$ means that the inputs are CIFAR backgrounds with objects pasted on them, with the task of classifying the inputs based on their CIFAR classes.

In total, we use four datasets which differ in their combinations of features $X$ and labels $Y$: the standard CIFAR-10 dataset ($X = C, Y = C$) the CIFAR-10 dataset with random objects pasted on it with the task of predicting CIFAR-10 classes ($X = C + O, Y = C$), the same augmented CIFAR-10 dataset with the task of predicting the category of the pasted objects ($X = C + O, Y = O$) and a dataset with only objects pasted on a black background, with the task of predicting the object category ($X = O, Y = O$). We use 10 object prototypes that are scaled down and pasted at a random position onto the CIFAR-10 images. Example images from these datasets can be found in Appendix A.2. We train models on each of the datasets, freeze their representations, and then fine-tune and evaluate each model's last layer on each of the other datasets. We use $X_p, Y_p$ to refer to a model's pretraining dataset and objective and $X_t, Y_t$ to refer to the dataset that its representations are transferred to.

| | | | Accuracy | | | | Sensitivity | |
|---|---|---|---|---|---|---|---|---|
| | X | | C | C + O | C + O | O | C + O | C + O |
| | | Y | C | C | O | O | C | O |
| Pre-training Dataset | C | C | $0.98_{\pm 0.00}$ | $0.89_{\pm 0.01}$ | $0.56_{\pm 0.06}$ | $1.00_{\pm 0.00}$ | $0.99_{\pm 0.00}$ | $0.21_{\pm 0.02}$ |
| | C + O | C | $0.98_{\pm 0.00}$ | $0.97_{\pm 0.00}$ | $0.15_{\pm 0.01}$ | $1.00_{\pm 0.00}$ | $1.00_{\pm 0.00}$ | $0.08_{\pm 0.01}$ |
| | C + O | O | $0.26_{\pm 0.02}$ | $0.13_{\pm 0.02}$ | $1.00_{\pm 0.00}$ | $0.98_{\pm 0.02}$ | $0.13_{\pm 0.02}$ | $0.99_{\pm 0.01}$ |
| | O | O | $0.29_{\pm 0.03}$ | $0.28_{\pm 0.03}$ | $0.25_{\pm 0.04}$ | $1.00_{\pm 0.00}$ | $1.00_{\pm 0.01}$ | $0.11_{\pm 0.04}$ |

Table 1: [**The impact of irrelevant features on downstream performance and invariance.**] Rows show pre-training tasks, with $X$ (*i.e.* $X_p$) denoting the features available in the training dataset and $Y$ (*i.e.* $Y_p$) the category of labels that the model was trained to predict. The accuracy columns denote the transfer accuracy after fine-tuning on the respective dataset ($X_t$) at predicting the respective label ($Y_t$). The sensitivity values show how sensitive resp. invariant the models' representations are according to the *sens*-metric defined in Section 3.4 on the $C+O$ data when changing the CIFAR background images ($Y_t = C$) while keeping the pasted foreground objects fixed, and while changing the pasted object ($Y_t = O$) while keeping the CIFAR background fixed. Lower values mean higher invariance. **Observations:** The representations of models pretrained with access to features that are irrelevant for their target task (objects $O$ for $X_p = C + O, Y_p = C$ and CIFAR backgrounds $C$ for $X_p = C + O, Y_p = O$) transfer worse to tasks where those features are important than their counterparts that did not have access to those features, *i.e.* $X_p = C, Y_p = C$ and $X_p = O, Y_p = O$. The sensitivity scores show that the difference in performance is due to representations becoming invariant to features that are not relevant to the pre-training objective, *i.e.* low sensitivity resp. high invariance for $X_p = C + O$ models towards the category $Y_t \neq Y_p$ they were not trained on, compared to high sensitivity towards the category $Y_t = Y_p$ they were trained on. Note that all models achieve $\sim 100\%$ accuracy on the $X_t = O, Y_t = O$ task, since they just have to separate 10 distinct object images.

| Model Type | Transformation Relationship | Synthetic Datasets | | | Real-World Datasets | |
|---|---|---|---|---|---|---|
| | | In-Distribution | Mild OOD | Strong OOD | CIFAR-10 | CIFAR-100 |
| **Image** | Same | $\mathbf{0.14_{\pm 0.07}}$ | $\mathbf{0.15_{\pm 0.07}}$ | $\mathbf{0.57_{\pm 0.16}}$ | $\mathbf{0.37_{\pm 0.21}}$ | $\mathbf{0.35_{\pm 0.18}}$ |
| | Other | $0.40_{\pm 0.15}$ | $0.39_{\pm 0.14}$ | $0.69_{\pm 0.15}$ | $0.50_{\pm 0.20}$ | $0.49_{\pm 0.18}$ |
| | None | $0.39_{\pm 0.15}$ | $0.42_{\pm 0.15}$ | $0.68_{\pm 0.16}$ | $0.44_{\pm 0.21}$ | $0.45_{\pm 0.19}$ |
| **Random** | Same | $\mathbf{0.12_{\pm 0.08}}$ | $\mathbf{0.44_{\pm 0.16}}$ | $\mathbf{0.24_{\pm 0.10}}$ | $\mathbf{0.39_{\pm 0.22}}$ | $\mathbf{0.36_{\pm 0.20}}$ |
| | Other | $0.64_{\pm 0.16}$ | $0.68_{\pm 0.16}$ | $0.33_{\pm 0.11}$ | $0.46_{\pm 0.21}$ | $0.45_{\pm 0.20}$ |
| | None | $0.65_{\pm 0.18}$ | $0.69_{\pm 0.17}$ | $0.35_{\pm 0.12}$ | $0.50_{\pm 0.20}$ | $0.49_{\pm 0.19}$ |

Table 2: [**Invariance transfer under distribution shift.**] Each cell shows the average *sens*-score (lower is more invariant), for models trained on image and random data, under distribution shift. "Transformation Relationship" refers to the relationship between the training and test transformations of the models. Rows labeled as "Same" show how invariant models are to their training transformations on each of the datasets (*e.g.* a translation-trained model evaluated on translated data), whereas rows labeled as "Other" show how invariant they are on average to transformations other than the ones they were trained on (*e.g.* a translation-trained model on rotations). "None" rows show the baseline invariance of models trained without transformations, *i.e.* simply to classify untransformed objects. The most invariant models in each category are shown in bold. Models are significantly more invariant to the transformations they were trained on than to other transformations, and this relationship persists on mild and strong OOD, as well as real-world data, indicating that a medium to high degree of representational invariance is preserved under distribution shifts.

**Results:** Table 1 shows the transfer accuracies of each model on each of the datasets. The main takeaway is that the transfer performance of the models differs significantly, depending on what information they have access to during pretraining, *i.e.* what features were *available*, and how *relevant* they are for the pretraining task. The impact that the relevance of input information has on transfer performance can be seen by comparing the two models trained on the augmented CIFAR images ($X_p = C + O$). The model pretrained to predict CIFAR labels ($Y_p = C$) performs well on this task ($Y_t = Y_p = C$), but poorly at predicting objects ($Y_t = O$), whereas the inverse relationship holds for its counterpart pretrained to predict objects ($Y_p = O$). The sensitivity scores (based on the *sens*-metric) show that that the representations of both models during pretraining become invariant to the irrelevant information in their inputs (objects or CIFAR backgrounds, respectively) and thus their representations cannot be used anymore to predict the corresponding classes ($Y_t \neq Y_p$). Additionally, models that had access to irrelevant features during pretraining ($X_p = C+O$) achieve lower transfer performance at predicting the category of classes they were not pretrained for ($Y_p \neq Y_t$) than models pretrained to predict the same category $Y_p$, but without access to the irrelevant features ($X_p = C$ and $X_p = O$).

The results show that during pretraining, the representations of models become invariant to information that is available in the input but irrelevant for the pretraining task. This effect can be exploited to harm transfer performance by introducing invariances to features that are relevant for downstream tasks into the representations. We further examine the relationship of relevance and availability with transfer performance in Appendix C.3. We find that more relevance gradually leads to less invariant representations, but that even if irrelevant objects are only available in a few inputs, models already become significantly more invariant to them.

**In summary,** our results show that invariance significantly affects how well representations transfer to downstream tasks. Its effect on transfer performance can be both positive, when representations share the right invariances with the target task, and negative, when they lack the right invariances or — even worse — possess invariance towards important information in the inputs.

## 5 How Transferable is Invariance?

So far, we have shown that sharing invariances between training and target tasks is quite important for the transfer performance of representations. But why does invariance have such an outsized influence? Our hypothesis is that invariances transfer well under distribution shift. If invariances learned on a pretraining

task are largely preserved in different domains, that would make them more robust than specific features that might change from domain to domain, and it might help to more robustly detect features that are present across domains.

### 5.1 Invariance Transfer Under Distribution Shift

To test how well invariance in pretrained representations transfers under distribution shift, we train models to be invariant to specific transformations using the Transforms-2D dataset and evaluate their invariance on out of distribution tasks. We create two categories of out of distribution (OOD) datasets: a mild OOD category with only small differences from the training dataset and a strong OOD category that is very different. Concretely, we create the mild OOD datasets by sampling a different set of image objects than the ones used for training. For the strong OOD datasets we use structured random data, by sampling a specific random pattern with pixel values distributed uniformly at random for each class and then pasting it on random background patterns, also sampled uniformly at random. Note that we can apply transformations to the random objects in the same way as to the image objects in Transforms-2D. Using this setup we train a different model for each of the 18 transformations in Transforms-2D, *i.e.* such that each of the models is invariant to a different transformation, and evaluate its invariance on each of the transformations, for each dataset category. We use ResNet-18 models here but report similar results for other architectures in Appendix D.2. To measure the invariance resp. sensitivity of a model's representation to a particular transformation, we use the *sens*-metric defined in Section 3.4.

We also study the reverse direction of invariance transfer, by training models on the random strong OOD data described above, and evaluating their performance on OOD data analogously but in reverse, *i.e.* mild OOD data uses a different set of random objects and strong OOD data is the Transforms-2D data for these models. In addition to the synthetic datasets, we also evaluate how well invariance transfers to real-world datasets, *i.e.* to CIFAR-10 and CIFAR-100, by using the Transforms-2D transformations as data augmentations. Additional details can be found in Appendix D.1.

**Results:** Table 2 shows the invariance of the two types of models on each of the dataset categories. We see that the invariance of models to the transformations that they were trained on is consistently higher than to other transformations on all datasets. Models trained to be invariant to the target transformations are also consistently more invariant than the baseline models trained without transformations across all the datasets. This shows that models do, in fact, acquire a significant degree of invariance to their training transformations, and are subsequently able to transfer it to other distributions. However, it seems to be more difficult to transfer invariance to random data (strong OOD for the image model and mild OOD for the random model) than to image data. It is also interesting to note that the image and random models achieve very similar degrees of invariance on the real-world datasets. This suggests that even training on structured random data can be a good prior for learning invariant representations, provided that the necessary transformations can be expressed on this type of data. Additional results and breakdowns of invariance over individual transformations can be found in Appendix D.2.

### 5.2 Invariance Mismatch between Training and Target Tasks

A different type of distribution shift happens when there is a mismatch in the transformations present in training compared to the target task, which we suspect often happens in real-world settings. To better understand these cases, we investigate the effect of training models on sub- and supersets of the transformations required by the target task. We create nested sets of transformations $T_i, i \in \{1, \ldots, 8\}$, such that $T_i \subset T_j$ for $i < j$ and $|T_i| = i$. For each $T_i$, we train a model (as described in Section 4.1) and then evaluate its transfer performance on data for each of the $T_i$'s.

**Results** in Figure 4 show that being invariant to more than the necessary transformations does not negatively impact performance (as long as those invariances do not conflict with the target task as in Section 4.2). However, if any required invariances are missing in the representations, a models' performance quickly decreases. This can help to explain why pretraining is often successful in practice: the invariances learned during pretraining do not need to perfectly match those required by the target task, as long as they cover a superset of required invariances. We hypothesize that commonly used pretraining datasets (such as Ima-

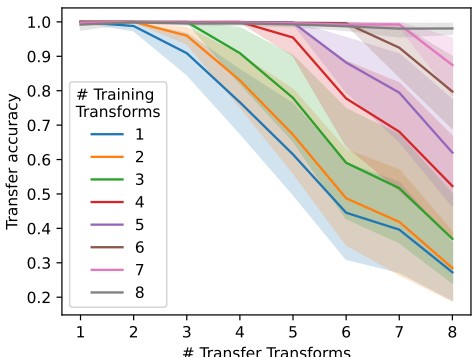

Figure 4: [**ResNet-18 models trained on nested sets of transformations and evaluated on datasets with super- and subsets of those transformations.**] Models trained on data with the same set or a superset of transformations as the target dataset consistently achieve almost 100% accuracy. However, models trained with only a subset of the transformations show considerably lower performance that decreases the smaller the subset of training transformations is compared to the target task. The results show that learning a superset of required invariances does not harm transfer performance but that missing required invariances degrades transfer performance.

geNet), together with data augmentations, induce a large set of broadly useful invariances. We show results for ResNet-18 models here and report very similar results for other architectures in Appendix D.3.

## 6 Conclusion

We study the importance of invariance in transfer learning by using a family of synthetic datasets, Transforms-2D, that allows us to precisely control differences between input points. By leveraging this method, we are able to show that invariance is a crucial factor in transfer learning and often as or more important than other factors such as the number of training samples, the model architecture and the class relationship of the training and target task. Sharing the right invariances with the target task positively impacts transfer performance, while a lack of the right invariance or even invariance towards important input features harms transfer performance. We further investigate the transferability of invariance under distribution shift and find that in most cases, models can transfer a high degree of invariance to new settings. Overall, our findings show that for transfer learning to be successful, the training and target tasks need to share important invariances.

**Limitations.** Since achieving precise control over data transformations is not possible with real-world data, our experiments heavily rely on synthetic data (see the discussion in Section 3.3). However, results derived from synthetic data might not generalize exactly to practical settings. To address this limitation we include validation experiments on real-world datasets in Sections 4.1 and 5.1 that show that the observations made using synthetic data can be largely extrapolated to more realistic settings as well.

**Broader Impact.** Our work primarily investigates invariance at a foundational level, and does not directly propose specific applications. However, we demonstrate in Section 4.2 that invariance can be exploited to harm transfer performance. An adversary might use techniques based on this insight to create models that do not easily transfer to particular downstream tasks. On the other hand, the same insight could also be used by model developers for alignment purposes, *i.e.* to make models less adaptable to certain harmful downstream applications.

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

# A  Dataset Details

## A.1  Transforms-2D Dataset Details

Transforms-2D datasets $\mathcal{D}(O, T)$ are parameterized by a set of foreground objects $O$ and a set of transformations of those objects $T$. An image is sampled by choosing a foreground object $o$ uniformly from a set of objects $O$, sampling a transformation $t$ from $T$ and pasting the transformed object $t(o)$ onto a background image chosen uniformly at random.

**Foreground objects and background images.** Foreground and background images are based on the SI-Score dataset Djolonga et al. (2021), which was introduced to study the connection between out-of-distribution and transfer performance. The creators of the SI-Score dataset have curated 61 foreground categories [1] (such as banana, jeans, sock, etc.) with a total of 614 object images, and 867 background images of nature scenes. The dataset is available under the following URL: `https://github.com/google-research/si-score`. It uses the Apache 2.0 license.

There are several images for each foreground category, *e.g.* several images of bananas, and each image has a transparency mask such that images only contain the object and no background. To be able to precisely control the variation between different images and to ensure that any changes in object appearance are only due to the transformations that are applied to them, we only use one image per foreground category throughout the analysis. In practice, we simply choose the image whose file name has the lowest lexicographical rank. For each parameterization $\mathcal{D}(O, T)$ of the Transforms-2D dataset, we always use the same set of 867 background images (hence they do not appear as a parameter), but choose a subset $O$ out of all available 61 objects categories.

**Transformations.** To cover a reasonably large variety of transformations and to ensure that our results are not overly dependent on the specific choice of transformations, we use three categories of 2D transformations: *geometric*, *photometric* and *corruption* transformations. Geometric transformations are affine transformations of the object geometry, photometric transformations change the color of the object and corruption transformations, inspired by Hendrycks & Dietterich (2019) degrade the quality the object. Each category consists of six transformation types, *e.g.* rotation is part of the geometric transformations, and for each transformation type, there can be potentially a large or infinite number of actual transformations (*e.g.* there are infinitely many rotations, one for each possible angle). Transformations are mainly implemented via standard PyTorch (Paszke et al., 2019) data augmentations. An overview over the transformations, their parameters and samples for each of them is shown in Figure 5.

We scale all images to a resolution of $32 \times 32$, which mimics the resolution of the CIFAR-10 and CIFAR-100 datasets Krizhevsky et al. (2009). Resolution is configurable though, and it is also possible to sample images with considerably higher resolution. We do not use additional data augmentations to train models, since that would interfere with the transformations present in the datasets. For most experiments, we use 50,000 training, 10,000 validation and 10,000 test samples, again mimicking the CIFAR datasets.

## A.2  Additional Information on the CIFAR-10 + Transforms-2D Dataset

Figure 6 shows example images for the augmented CIFAR-10 images that we use in the analysis in section 4.2 to investigate the effects of irrelevant information on learned invariances.

---

[1] The original paper Djolonga et al. (2021) mentions 62 categories, but publicly available dataset only contains 61 categories.

| Category | Transformation | Parameters used | Samples |
|---|---|---|---|
| Geometric | Translate | x and y position by $[0\%, 50\%]$ image size | |
| | Rotate | by $[0, 360]$ degrees | |
| | Scale | to $[40\%, 100\%]$ size | |
| | Shear | x and y by $[-50, 50]$ degrees | |
| | Vertical flip | with 50% probability | |
| | Horizontal flip | with 50% probability | |
| Photometric | Hue | deviation in $[-0.5, 0.5]$ | |
| | Brightness | change in $[-1, 1]$ | |
| | Grayscale | with 50% probability | |
| | Posterize | with 50% probability to 1 bit | |
| | Invert | with 50% probability | |
| | Sharpen | with probabiilty 50%, sharpness factor 7 | |
| Corruption | Gaussian blur | kernel size 7 pixels, $\sigma \in [0.1, 1.5]$ | |
| | Gaussian noise | with $\mu = 0, \sigma = 1$, probability 50% | |
| | Pixelate | to half resolution, probability 50% | |
| | Elastic distortion | $\alpha = 150$, probability 50% | |
| | Erasing | square with 14 x 14 pixels size at random position, probability 50% | |
| | Contrast | change in $[-1, 1]$ | |

Figure 5: [**Transforms-2D transformations.**] Categories, transformation types, transformation parameters and samples generated using the transformations for the Transforms-2D dataset.

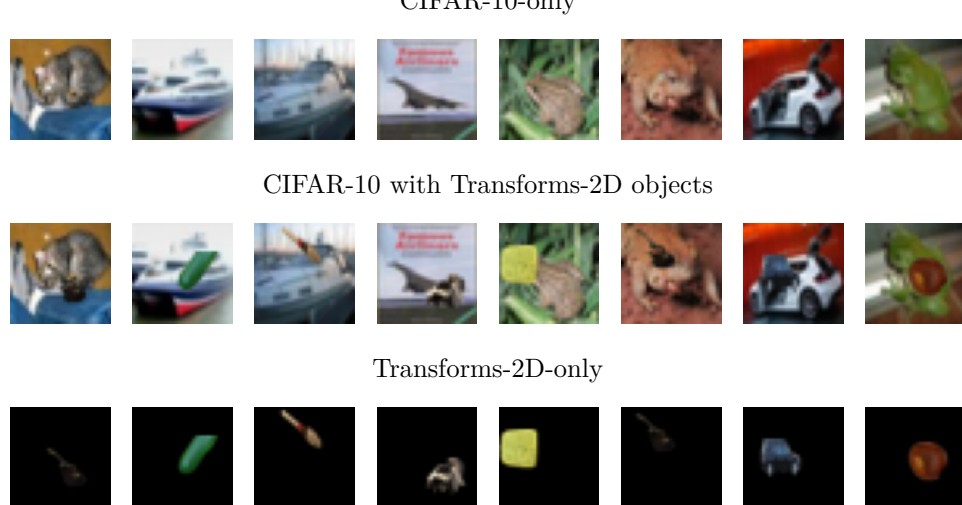

Figure 6: [**Examples of test images in the irrelevant feature analysis**]. From top to bottom: CIFAR-10 only ($X = C$), CIFAR-10 with pasted objects ($X = C + O$), objects only ($X = O$)

## B    Additional details on the training and evaluation setup

### B.1    Architectures

For most of the experiments in the paper we use ResNet-18 models (He et al., 2016). They are generally sufficient for the 32 x 32 input size that we use for the Transforms-2D dataset and achieve close to 100% accuracy on their training distributions.

We also compare the importance of invariance with other pretraining factors, including architectures in Section 4.1 and in Appendix C.1. For this, we additionally use ResNet-50 (He et al., 2016), DenseNet-121 Huang et al. (2017), VGG-11 Simonyan & Zisserman (2014) and Vision Transformer (ViT) Dosovitskiy et al. (2020) models. All models are implemented in PyTorch (Paszke et al., 2019). For all CNN models we use implementations adapted for CIFAR datasets available here `https://github.com/kuangliu/pytorch-cifar`. For ViTs, we use an implementation from this repository `https://github.com/omihub777/ViT-CIFAR` (achieving more than 80% accuracy on CIFAR-10) with a patch size of 8, 7 layers, 384 hidden and MLP units, 8 heads and no Dropout.

### B.2    Training

We train models using Pytorch Lightning (Falcon & The PyTorch Lightning team, 2019) on the Transforms-2D dataset for 50 epochs and fine-tune their output layer (while keeping the rest of the network frozen) for 200 epochs. We find that 50 training epochs are sufficient for models to reach close to 100% accuracy. For fine-tuning we choose a larger number of 200 epochs, because models that have been pretrained with invariances different from those required by the target task typically converge only slowly. Models that have been trained with the same invariances on the other hand converge much faster.

All CNN models are trained and fine-tuned using the Adam optimizer (Kingma & Ba, 2014) with a learning rate of 0.001. For ViTs, we use cosine learning rate scheduler (Loshchilov & Hutter, 2016) with a learning rate that is decayed from 0.001 to 0.00001 over the duration of training, and a weight decay of 0.00001, with 5 warmup epochs. We keep the checkpoint that achieves the highest validation accuracy during training and fine-tuning.

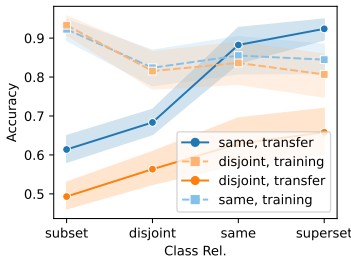

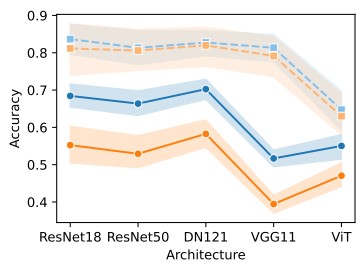

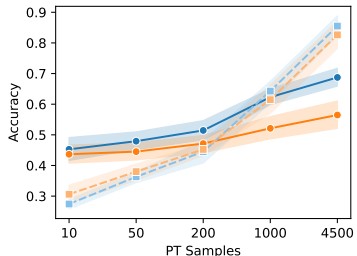

(a) Invariance vs class relationship.
(b) Invariance vs architecture.
(c) Invariance vs number of training samples per class.

Figure 7: [**Invariance to data augmentations vs the importance of other factors on CIFAR-10.**] Training and transfer performance for models trained with different factors of variation and different transformations on the CIFAR-10 dataset.

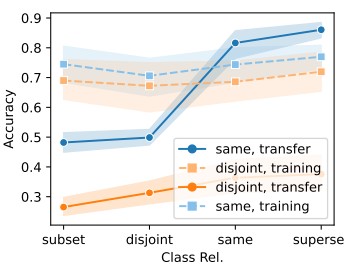

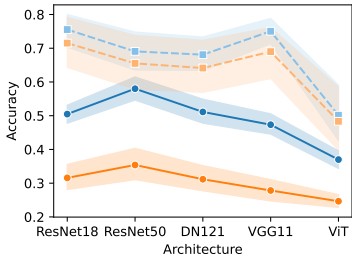

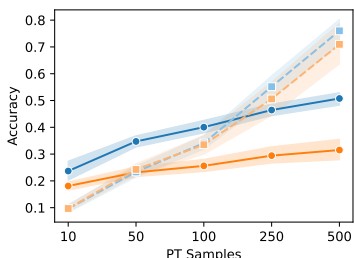

(a) Invariance vs class relationship.
(b) Invariance vs architecture.
(c) Invariance vs number of training samples per class.

Figure 8: [**Invariance to data augmentations vs the importance of other factors on CIFAR-100.**] Training and transfer performance for models trained with different factors of variation and different transformations on the CIFAR-100 dataset.

## B.3    Experiments

We repeat each experiment 10 times and report mean and variance values. Shaded regions in the plots indicate 95% confidence intervals, and annotations in the tables indicate one standard deviation. During each run, we randomize the configuration of the Transforms-2D dataset and the sampling process. For the configuration, this means that we sample different sets of objects $O$ and different sets of transformation types in $T$. For the sampling process, we randomize the order of sampling foreground objects, transformations of each transformation type and background images.

## C    Additional Results on the Importance of Invariance for Transfer Performance

### C.1    Real-world Experiments on Augmented CIFAR data

We conduct experiments analogous to those described in Section 4.1 on real-world CIFAR-10 and CIFAR-100 data. We observe results similar to those reported in Figure 2 for CIFAR-10 in Figure 7 and for CIFAR-100 in Figure 8. In all cases, the model trained with the same transformations as the target task significantly outperforms its counterpart pretrained with a disjoint set of transformations (in most cases between 10% and 20%). One difference to the results reported in Section 4.1 is that in Figure 2, even the worst performing model using the same transformations as the target task outperforms the best-performing model using a disjoint set of transformations. This is no longer the case with the CIFAR models, for example the model with disjoint transformations trained with 4500 samples per class outperforms the model with the same

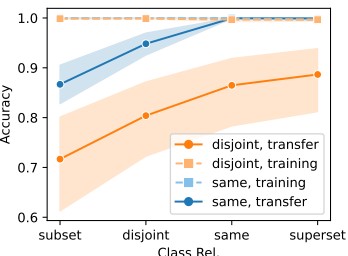 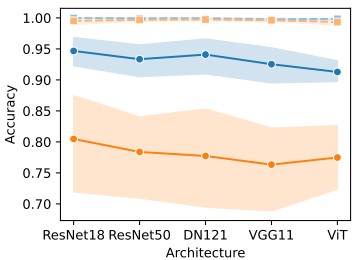 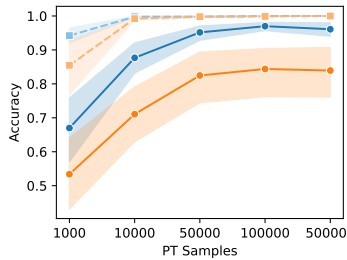

(a) Invariance vs class relationship.    (b) Invariance vs architecture.    (c) Invariance vs number of pretraining samples.

Figure 9: [**Invariance to data augmentations vs the importance of other factors on Transforms-2D with full fine-tuning on 200 samples.**] Training and transfer performance for models trained with different factors of variation and different transformations on the Transforms-2D dataset, fine-tuned on 200 samples after linear probing.

transformations trained with 200 samples per class (but not the one with 1000 samples per class). This is somewhat expected, though, as more samples presumably help the models to better learn the invariances inherent in the different classes in the dataset, as opposed to transformation-induced invariances.

**Takeaways:** The results on real-world CIFAR data largely confirm our findings made using the synthetic Transforms-2D dataset. Invariance to the right transformations significantly improves the transfer performance of models and is a very important factor compared to other pretraining dimensions such as the number of samples, the architecture and the class relationship. E.g. on CIFAR-10, you need roughly 20 times more samples to compensate for a mismatch in invariance.

## C.2    How Does Fine-tuning the Whole Model Impact Invariance?

We want to better contextualize our findings and understand how difficult it is to change the invariance properties of pretrained representations. Therefore, we apply an additional fine-tuning pass to the models in Section 4.1 and Appendix C.1 on the target dataset after their last layer has been tuned on top of a frozen feature extractor, mimicking the linear probing, then full fine-tuning approach discussed in Kumar et al. (2022).

We fine-tune models after they have been trained with linear probing on the target task for and additional 50 epochs with a learning rate of $10^{-3}$ (chosen based on a grid search over the values $10^{-1}, 10^{-2}, 10^{-3}, 10^{-4}, 10^{-5}$) and a linearly decreasing learning rate schedule. For Transforms-2D, we explore two fine-tuning dataset sizes: a low-data setting with 200 and a high-data setting with 2000 fine-tuning samples. For the augmented CIFAR-10 and CIFAR-100 datasets, we use 1% and 10% of the original training data for the low- and high-data settings, respectively. Again, results are averages over 10 runs.

Figure 9 shows the results for Transforms-2D with 200 samples, Figure 10 for CIFAR-10 with 1% of the full training samples and Figure 11 for CIFAR-100 with 1% of the full training samples. Figure 12 and Figure 13 show the comparison of the differences in transfer performance due to invariance vs other factors, for the low- and high-data fine-tuning scenarios, respectively. The results show that full fine-tuning can change the invariance properties of representations such that even representations not trained with the right set of invariances can adapt to the target task. However, the degree to which this is possible depends on the amount of available fine-tuning data. Especially for Transforms-2D in the low-data setting, the representations trained with the right invariances still perform significantly better than the ones trained with disjoint transformations. In the high-data setting the difference diminishes, since the training starts to approximate full re-training on the target task. However, the representations trained with the same transformations as the target task still perform at least as well as those trained with different transformations, in all cases.

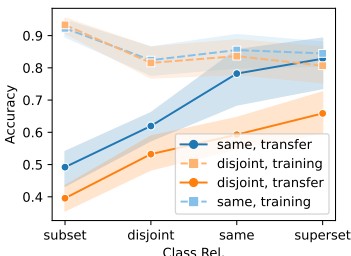
(a) Invariance vs class relationship.

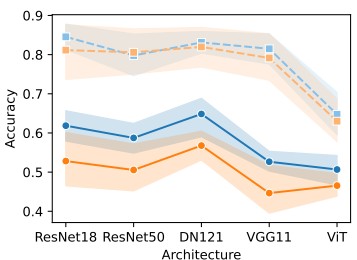
(b) Invariance vs architecture.

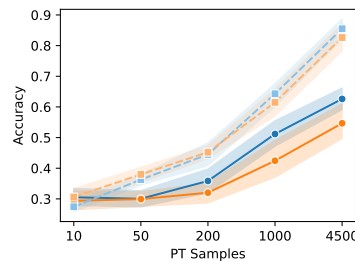
(c) Invariance vs number of pretraining samples.

Figure 10: [**Invariance to data augmentations vs the importance of other factors on augmented CIFAR-10 with full fine-tuning on 1% of the full training samples.**] Training and transfer performance for models trained with different factors of variation and different transformations on the CIFAR-10 dataset, fine-tuned on 1% of the full training set samples after linear probing.

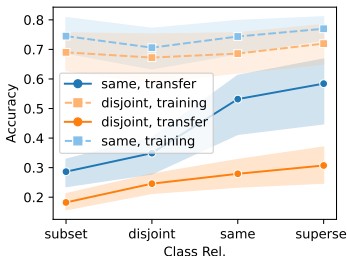
(a) Invariance vs class relationship.

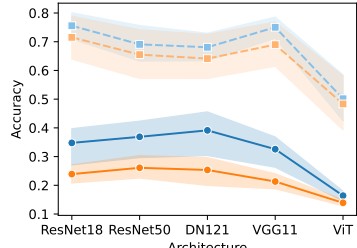
(b) Invariance vs architecture.

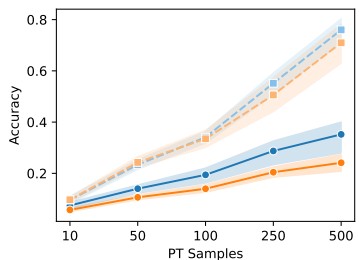
(c) Invariance vs number of pretraining samples.

Figure 11: [**Invariance to data augmentations vs the importance of other factors on augmented CIFAR-100 with full fine-tuning on 1% of the full training samples.**] Training and transfer performance for models trained with different factors of variation and different transformations on the CIFAR-100 dataset, fine-tuned on 1% of the full training set samples after linear probing.

### C.3 How Does Relevance and Availability of Input Information Impact Invariance?

In Section 4.2 we have shown that the *relevance* and *availability* of input information impacts the invariances learned by a model. In this Section, we make two observations.

First, we show that the two models trained on CIFAR-10 images with objects pasted on them $(X_p = C + O)$ perform very differently after fine-tuning at predicting CIFAR-10 classes $(Y_t = C)$ and object categories $(Y_t = O)$ respectively, depending on whether they were pre-trained to predict the CIFAR-10 classes $(Y_p = C)$ or object categories $(Y_p = O)$, even though they saw exactly the same input data. This result shows that the relevance of a feature for a target task is a key driver in determining which input features a model becomes invariant to.

Second, the models trained on only CIFAR-10 images to predict CIFAR-10 classes $(X_p = C, Y_p = C)$ and on only object images to predict object categories $(X_p = O, Y_p = O)$ show better transfer performance on the respective other task $(Y_t \neq Y_p)$ than their counterparts that were trained on CIFAR-10 images with pasted objects $(X_p = C + O)$. These two models did not have access to the object- and CIFAR-features respectively and could therefore not develop invariances towards them, as did the models that saw them during training. This means that another important property in determining the invariances learned by a model is the availability of features during training. Next, we investigate the effects of relevance and availability in more detail.

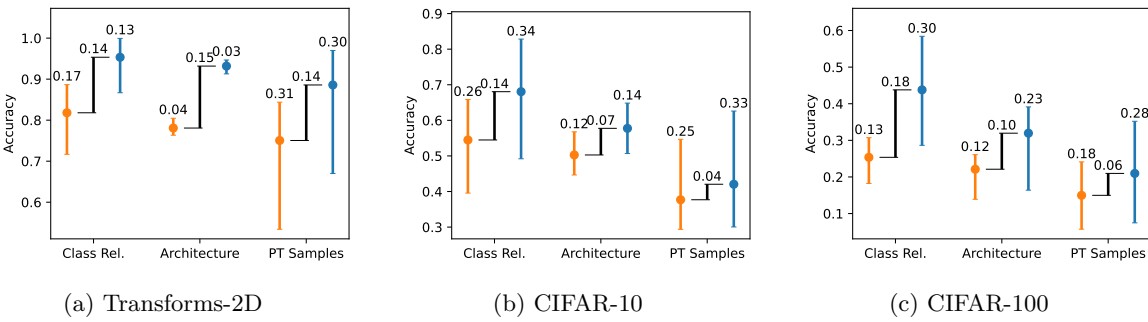

(a) Transforms-2D        (b) CIFAR-10        (c) CIFAR-100

Figure 12: [**Difference in transfer performance due to invariance vs other factors, for full fine-tuning, for 200 samples on Transforms-2D and 1% of samples on the CIFAR datasets.**] We compare the differences in transfer performance caused by representational invariance with the differences caused by changes to other factors on the Transforms-2D and the CIFAR-10 and CIFAR-100 datasets (with data augmentations). Orange (blue) bars show the span of transfer performance for different-transformation models $g_d$ (same-transformation models $g_s$), for each comparison factor (class relationship, architecture, number of samples). Black bars show the difference between transfer performance means across factor values for $g_d$ and $g_s$, *i.e.* the difference in performance caused by having the same vs different invariances as the target task.

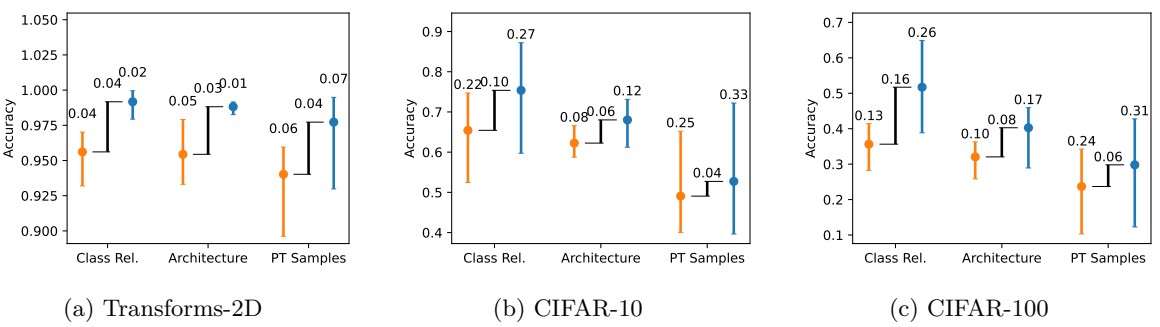

(a) Transforms-2D        (b) CIFAR-10        (c) CIFAR-100

Figure 13: [**Difference in transfer performance due to invariance vs other factors, for full fine-tuning, for 2000 samples on Transforms-2D and 10% of samples on the CIFAR datasets.**] We compare the differences in transfer performance caused by representational invariance with the differences caused by changes to other factors on the Transforms-2D and the CIFAR-10 and CIFAR-100 datasets (with data augmentations). Orange (blue) bars show the span of transfer performance for different-transformation models $g_d$ (same-transformation models $g_s$), for each comparison factor (class relationship, architecture, number of samples). Black bars show the difference between transfer performance means across factor values for $g_d$ and $g_s$, *i.e.* the difference in performance caused by having the same vs different invariances as the target task.

**Relevance.** To understand how the invariance to features changes with their relevance for the target task, we train models on the dataset described in Section 4.2 and Appendix A.2. However, we now introduce correlation of different strengths between the objects and the CIFAR labels. In particular, we train models $g_\alpha$ on datasets $\mathcal{D}_\alpha$, where $\alpha \in \{0, 0.2, 0.4, 0.6, 0.8, 0.85, 0.9, 0.95, 1.0\}$ is the correlation strength between CIFAR labels and object categories. Input images in $\mathcal{D}_\alpha$ are constructed by pasting one specific object per CIFAR-10 class (out of a set of 10 total objects) on the CIFAR-10 training images $\alpha$-fraction of the time and pasting a random object otherwise. Then, we fine-tune the $g_\alpha$s' last layers and evaluate them on the augmented CIFAR-10 images, both for predicting the CIFAR-10 class ($X_t = C + O, Y_t = C$) as well as the object category ($X_t = C + O, Y_t = O$).

**Availability.** To investigate the effect of availability, we train models on datasets $\mathcal{D}_\beta, \beta \in \{0, 0.2, 0.4, 0.6, 0.8, 1.0\}$ that are equivalent to the $X = C + O, Y = C$ dataset, *i.e.* objects are pasted

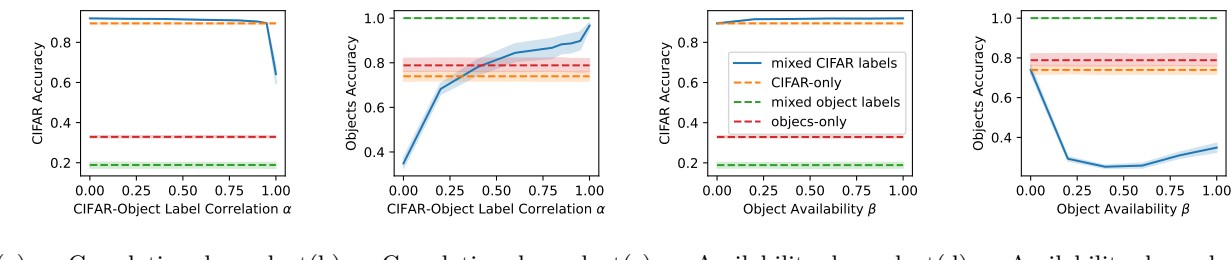

(a) Correlation-dependent transfer performance to CIFAR-10 labels

(b) Correlation-dependent transfer performance to object labels

(c) Availability-dependent transfer performance to CIFAR-10 labels

(d) Availability-dependent transfer performance to object labels

Figure 14: [**The effect of feature relevance and availability on learned invariances**]
Left plots: Transfer accuracy on the $X_t = C+O$ dataset for models trained with different correlation strengths of pasted object categories with CIFAR labels. As the correlation resp. relevance of the pasted objects for the target task increases, models become increasingly better at predicting them and their representations become more sensitive to their presence.
Right plots: Transfer accuracy on the $X_t = C + O$ dataset for models trained with different availability of pasted object categories. As soon as a feature becomes available in the input but is irrelevant for the target task, models start to become invariant to it.

randomly onto CIFAR backgrounds, but with the difference that objects are only pasted $\beta$-fraction of the time for dataset $\mathcal{D}_\beta$. We again fine-tune and evaluate the last layer of the models trained on $\mathcal{D}_\beta$ on the same datasets as for the relevance analysis. In both cases we paste the object in the upper right corner of the image.

Figure 14a and Figure 14b show the transfer accuracies of models trained to predict CIFAR-10 classes and object categories, respectively. The blue curve correspond to the models pretrained on $X_p = C + O$ datasets where the objects were pasted with different correlations with the CIFAR labels. Dashed lines show the performance of reference models. As the correlation of the pasted object categories with the CIFAR-10 classes increases, CIFAR-10 transfer accuracy mostly remains constant, whereas object category accuracy increases steadily. Only when pasted objects and CIFAR labels become perfectly correlated, CIFAR-10 accuracy drops and object detection accuracy reaches 100%. This trend shows that as the pasted objects become more relevant for the target task, the models' representations gradually become less invariant to them and allow for increasingly better prediction of the object category. CIFAR-10 accuracy on the other hand remains constant, up to the point where the pasted object can reliably replace the CIFAR background.

The CIFAR performance does not depend on the availability of the pasted objects, *i.e.* whether or not an irrelevant object is present in the training data has no impact on it. Object category prediction performance on the other hand quickly drops as soon as the model has access to the irrelevant object features, showing that it immediately develops an invariance towards the objects. In summary, relevance and availability both play a role in determining which invariances a model learns, but relevance seems to be the more important property.

# D    Additional Details and Results on Invariance Transfer

## D.1    Details on the invariance transfer experiments

**Data.**    For the synthetic image datasets we use the Transforms-2D dataset and generate samples as described in Appendix A.1. We create out of distribution variants by using a new set of image objects, that the models have not seen during training. We create the synthetic random datasets by sampling pixel-values for foregrounds and backgrounds uniformly at random. Classes are created by using a different fixed random foreground pattern for each class, *i.e.* all samples for a class are transformed versions of the same random pattern on different random backgrounds. Random patterns are square-shaped, and have a length of 70% of the image size along each dimension. Examples of random images can be seen in Figure 15a. For both

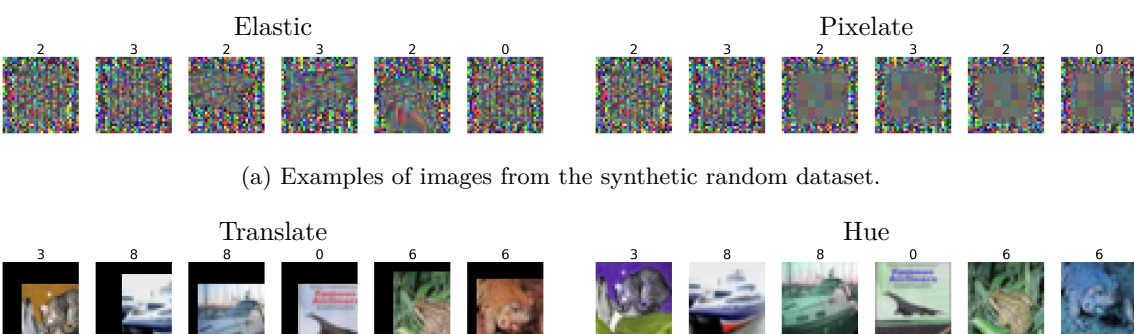

(a) Examples of images from the synthetic random dataset.

(b) Examples of augmented CIFAR-10 images.

Figure 15: [**Examples of images used for the OOD analysis.**]. The top part shows examples of images from the synthetic random dataset, while the bottom part shows examples of augmented CIFAR-10 images. Each row shows the examples for a single transformation.

| Model Type | Transformation Relationship | Synthetic Datasets | | | Real-World Datasets | |
|---|---|---|---|---|---|---|
| | | In-Distribution | Mild OOD | Strong OOD | CIFAR-10 | CIFAR-100 |
| **Image** | Same | $\mathbf{0.12}_{\pm 0.06}$ | $\mathbf{0.19}_{\pm 0.09}$ | $\mathbf{0.66}_{\pm 0.16}$ | $\mathbf{0.34}_{\pm 0.17}$ | $\mathbf{0.33}_{\pm 0.16}$ |
| | Other | $0.39_{\pm 0.12}$ | $0.39_{\pm 0.12}$ | $0.75_{\pm 0.15}$ | $0.50_{\pm 0.17}$ | $0.49_{\pm 0.16}$ |
| | None | $0.38_{\pm 0.12}$ | $0.39_{\pm 0.12}$ | $0.78_{\pm 0.16}$ | $0.51_{\pm 0.18}$ | $0.49_{\pm 0.17}$ |
| **Random** | Same | $\mathbf{0.12}_{\pm 0.08}$ | $\mathbf{0.41}_{\pm 0.17}$ | $\mathbf{0.20}_{\pm 0.12}$ | $\mathbf{0.33}_{\pm 0.20}$ | $\mathbf{0.30}_{\pm 0.19}$ |
| | Other | $0.64_{\pm 0.13}$ | $0.71_{\pm 0.14}$ | $0.29_{\pm 0.11}$ | $0.44_{\pm 0.16}$ | $0.43_{\pm 0.16}$ |
| | None | $0.65_{\pm 0.13}$ | $0.73_{\pm 0.15}$ | $0.26_{\pm 0.13}$ | $0.41_{\pm 0.19}$ | $0.40_{\pm 0.18}$ |

Table 3: [**Invariance transfer under distribution shift for VGG-11 models.**]

synthetic image and random datasets, we use 30 classes and a single type of transformation per dataset. Images have a size of 32x32 pixels.

For the real world datasets (CIFAR-10 and CIFAR-100 Krizhevsky et al. (2009)), we apply the transformations from the Transforms-2D dataset as data augmentations. We use the same transformations as in the synthetic datasets, with the exception of the translation transformation, which here translates images by up to 30% of the image size along each axis, instead of positioning objects at a random position in the image. This change is necessary since with the CIFAR images occupying the entire image, translations would otherwise have no effect. Examples of augmented CIFAR-10 images can be seen in Figure 15b.

**Measuring invariance.** For the experiments here, we report the average *sens*-score as defined in Section 3.4, *i.e.* the ratio of L2-distance between representations at the penultimate layer for inputs that only differ in their transformations, to the L2-distance between any two inputs from the respective dataset (lower values mean more invariance). For example, when computing the *sens*-score for the translation transformation, we compute the average L2-distances between instances of the same object on the same background in different positions, divided by the average L2-distances of different objects on different backgrounds, in different positions. We use the penultimate layer representations here, since those are the representations most relevant for transfer learning.

## D.2 Additional results on invariance transfer

**Invariance transfer results for additional architectures.** Tables 3, 4 and 5 show the results for the experiments on invariance transfer under distribution shift for VGG-11, DenseNet-121 and ViT models, respectively. Each cell shows the average *sens*-score (lower values mean more invariance), for models trained on image and random data. "Transformation Relationship" refers to the relationship between the training and test transformations of the models. Rows labeled as "Same" show how invariant models are to their

| Model Type | Transformation Relationship | Synthetic Datasets | | | Real-World Datasets | |
|---|---|---|---|---|---|---|
| | | In-Distribution | Mild OOD | Strong OOD | CIFAR-10 | CIFAR-100 |
| **Image** | Same | $\mathbf{0.16_{\pm 0.09}}$ | $\mathbf{0.24_{\pm 0.10}}$ | $\mathbf{0.67_{\pm 0.14}}$ | $\mathbf{0.42_{\pm 0.20}}$ | $\mathbf{0.40_{\pm 0.18}}$ |
| | Other | $0.39_{\pm 0.12}$ | $0.40_{\pm 0.12}$ | $0.73_{\pm 0.14}$ | $0.50_{\pm 0.18}$ | $0.48_{\pm 0.17}$ |
| | None | $0.39_{\pm 0.13}$ | $0.41_{\pm 0.13}$ | $0.74_{\pm 0.15}$ | $0.50_{\pm 0.19}$ | $0.48_{\pm 0.17}$ |
| **Random** | Same | $\mathbf{0.17_{\pm 0.10}}$ | $\mathbf{0.50_{\pm 0.12}}$ | $\mathbf{0.24_{\pm 0.09}}$ | $\mathbf{0.44_{\pm 0.22}}$ | $\mathbf{0.42_{\pm 0.21}}$ |
| | Other | $0.66_{\pm 0.15}$ | $0.72_{\pm 0.14}$ | $0.28_{\pm 0.10}$ | $0.46_{\pm 0.22}$ | $0.44_{\pm 0.20}$ |
| | None | $0.69_{\pm 0.17}$ | $0.75_{\pm 0.14}$ | $0.30_{\pm 0.10}$ | $0.46_{\pm 0.22}$ | $0.46_{\pm 0.21}$ |

Table 4: [**Invariance transfer under distribution shift for DenseNet-121 models.**]

| Model Type | Transformation Relationship | Synthetic Datasets | | | Real-World Datasets | |
|---|---|---|---|---|---|---|
| | | In-Distribution | Mild OOD | Strong OOD | CIFAR-10 | CIFAR-100 |
| **Image** | Same | $\mathbf{0.22_{\pm 0.09}}$ | $\mathbf{0.26_{\pm 0.11}}$ | $\mathbf{0.52_{\pm 0.18}}$ | $\mathbf{0.36_{\pm 0.14}}$ | $\mathbf{0.33_{\pm 0.13}}$ |
| | Other | $0.41_{\pm 0.17}$ | $0.41_{\pm 0.16}$ | $0.64_{\pm 0.19}$ | $0.48_{\pm 0.17}$ | $0.46_{\pm 0.17}$ |
| | None | $0.43_{\pm 0.19}$ | $0.42_{\pm 0.17}$ | $0.65_{\pm 0.19}$ | $0.48_{\pm 0.18}$ | $0.47_{\pm 0.18}$ |
| **Random** | Same | $\mathbf{0.28_{\pm 0.13}}$ | $\mathbf{0.43_{\pm 0.15}}$ | $\mathbf{0.23_{\pm 0.11}}$ | $\mathbf{0.31_{\pm 0.14}}$ | $\mathbf{0.30_{\pm 0.14}}$ |
| | Other | $0.61_{\pm 0.20}$ | $0.64_{\pm 0.19}$ | $0.34_{\pm 0.14}$ | $0.45_{\pm 0.18}$ | $0.43_{\pm 0.18}$ |
| | None | $0.63_{\pm 0.20}$ | $0.67_{\pm 0.19}$ | $0.34_{\pm 0.15}$ | $0.46_{\pm 0.19}$ | $0.45_{\pm 0.19}$ |

Table 5: [**Invariance transfer under distribution shift for ViT models.**]

training transformations on each of the datasets (*e.g.* a translation-trained model evaluated on translated data), whereas rows labeled as "Other" show how invariant they are to transformations other than the ones they were trained on (*e.g.* a translation-trained model on rotations). "None" rows show the baseline invariance of models trained without transformations, *i.e.* simply to classify untransformed objects. The most invariant models in each category are shown in bold. Results are computed over 10 runs that randomize the image and random objects, backgrounds, the way transformation values are sampled and the weight initialization of the models. The subscript numbers indicate the standard deviation of the *sens*-score over the 10 runs.

For all architectures, we observe very similar trends, which indicates that our observations about how models transfer invariance are robust. First, training models to be invariant to specific transformations indeed makes them more invariant to primarily those transformations. The invariance on the training transformations ("Same" rows) is significantly higher than for other transformations ("Other" rows), and also higher than the invariance of the baseline models trained without transformations ("None") rows.

Second, models can transfer a medium to high degree of invariance to out-of-distribution data. The invariance of models trained on image data is similar to that on other image data with different foreground objects ("Mild OOD"), and the same is roughly true for models trained on random data when transferring to image data ("Strong OOD"). Both types of models are less invariant on (unseen) random data ("Strong OOD" for the image models, "Mild OOD" for the random models), but they are still more invariant to their training transformations than to other transformations, and also more invariant than the baseline models trained without transformations. Both types of models achieve high invariance on the real-world CIFAR-10 and CIFAR-100 datasets. These results indicate that models can retain a high degree of invariance, even when the data that they are evaluated on is substantially different from the data they were trained on. This helps to explain why invariance is such an important property for transfer learning: it can be transferred well to out-of-distribution data.

Third, the models trained on image and random data achieve similar invariance on the real-world CIFAR datasets. This indicates that for learning representations that are invariant to certain transformations, the specific features of the data do not matter as much. The caveat is that this approach only works if the features in the dataset are suitable to express the desired transformations. The pixel-level transformations in the Transforms-2D datatset can be applied to random data, but for higher-level transformations, *e.g.* transfor-

mations of the pose of an animal, the dataset needs to contain features that can express the transformation, *e.g.* animals with certain poses.

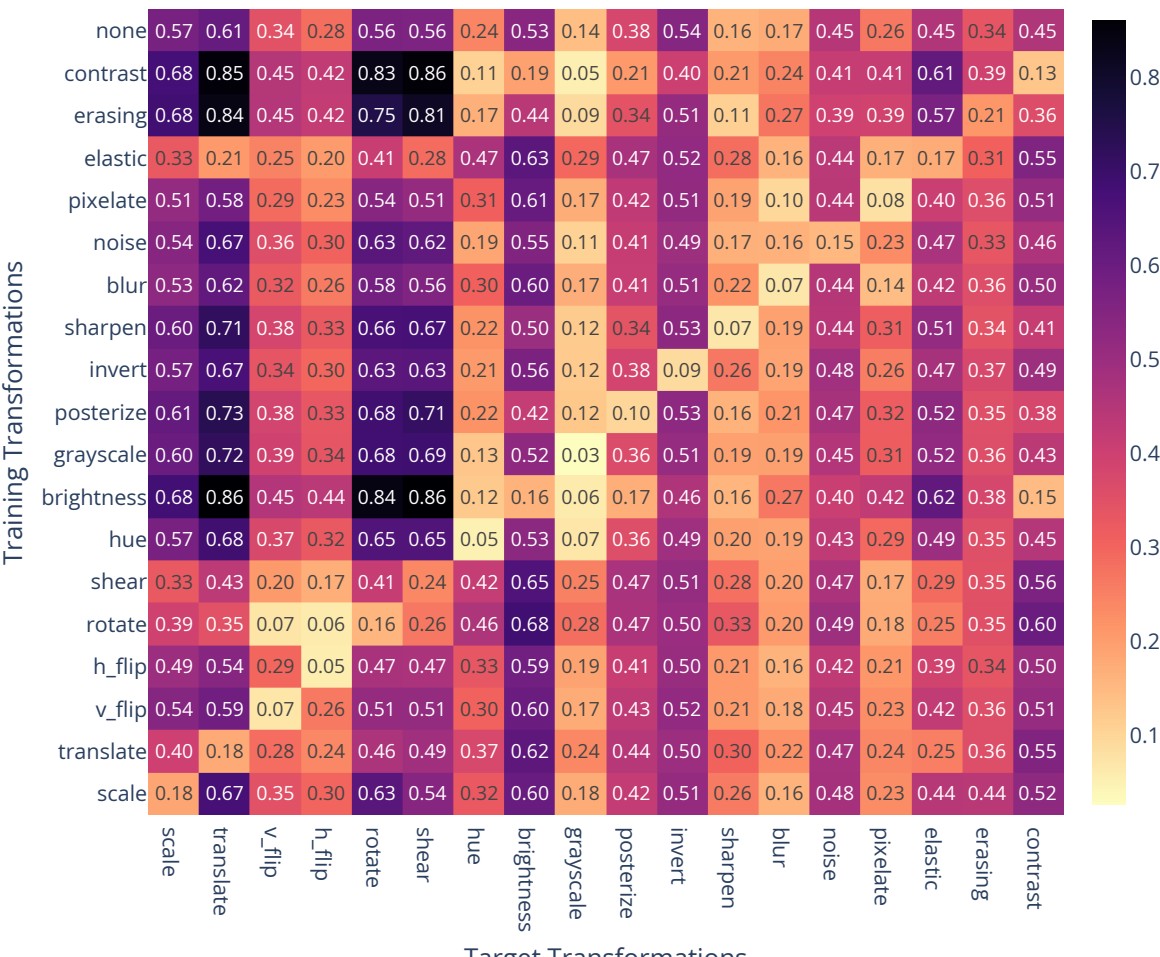

Figure 16: [**Invariance transfer between transformations, for models trained on image data.**] Rows show how invariant models trained to be invariant to a specific transformation are to each of the transformations. Columns show how invariant each model is to the specific target transformation. Values are computed using the *sens*-metric (see Section 3.4). Lower numbers indicate higher invariance. Models trained with a specific transformation in their training data become more invariant to this transformation. We show the invariance of each model on the test set of its training dataset type ("In-Distribution").

**Invariance Transfer Results Between Different Transformations.** We show a breakdown of the invariance transfer on a per-transformation basis, on the training distributions (objects and backgrounds) of the image and random models for ResNet-18 architectures in Figures 16 and 17, respectively. For each transformation, we show the invariance of models trained to be invariant to that transformation, *i.e.* trained to classify objects modified by that transformation, for each of the other transformations. The results show that models indeed become primarily invariant to their training transformations (low values on the diagonal). However, we can also observe some "spillovers", *e.g.* training models to be rotation- and shear-invariant also increases the invariance to horizontal and vertical flips (but not vice versa), and training for hue- and grayscale-invariance increases the invariance to the respective other transformation.

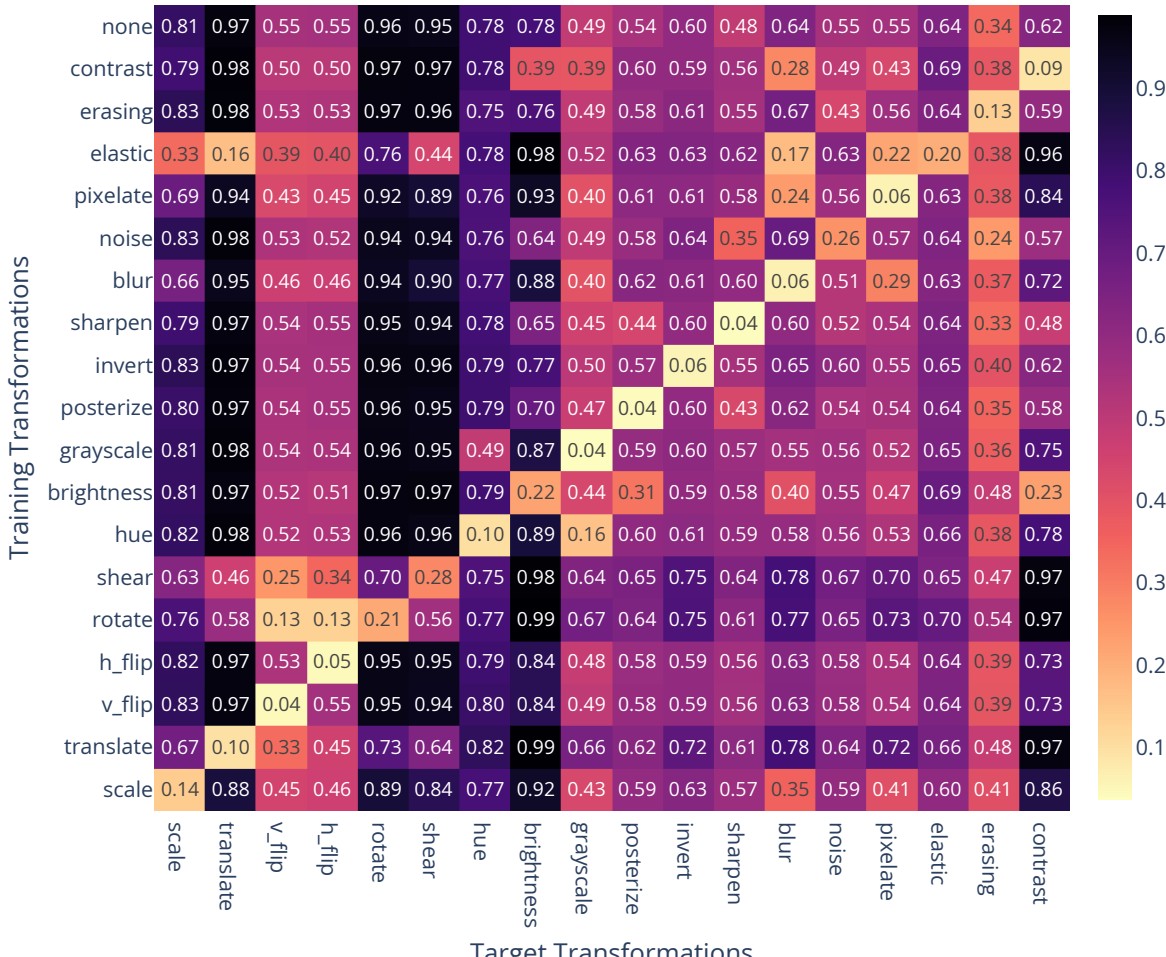

Figure 17: [**Invariance transfer between transformations, for models trained on random data.**] Rows show how invariant models trained to be invariant to a specific transformation are to each of the transformations. Columns show how invariant each model is to the specific target transformation. Values are computed using the *sens*-metric (see Section 3.4). Lower numbers indicate higher invariance. Models trained with a specific transformation in their training data become more invariant to this transformation. We show the invariance of each model on the test set of its training dataset type ("In-Distribution").

### D.3    Additional Results for Invariance Mismatch

Figure 18 shows the results analogous to those in Section 5.2 for invariance mismatch between training and transfer tasks, for addtional model families: VGG-11, DenseNet-121 and ViT models. We observe the same pattern as in Section 5.2, *i.e.* models trained to be invariant to the same set or a superset of transformations as those in the target dataset consistently achieve almost 100% accuracy, but models that are missing invariance to transformations in the target dataset achieve significantly lower performance.

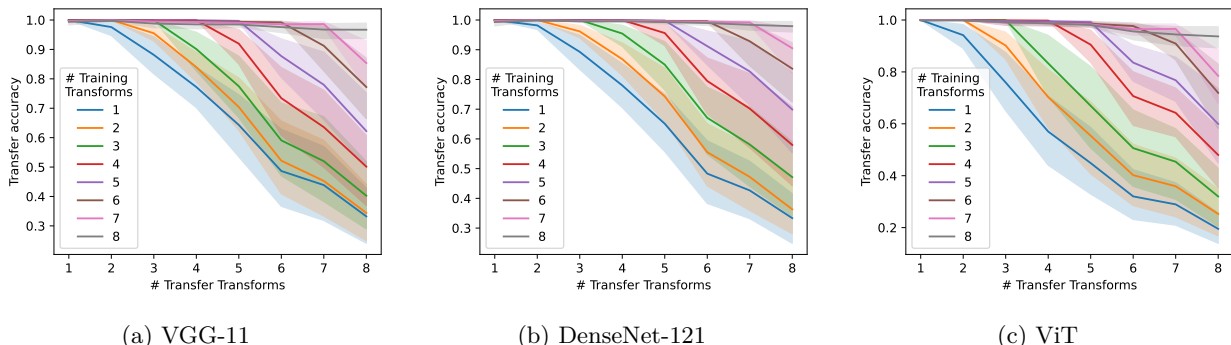

(a) VGG-11        (b) DenseNet-121        (c) ViT

Figure 18: [**Models trained on nested sets of transformations and evaluated on datasets with super- and subsets of those transformations.**]. Models trained on data with the same set or a superset of transformations as the target dataset consistently achieve almost 100% accuracy. However, models trained with only a subset of the transformations show considerably lower performance that decreases the smaller the subset of training transformations is compared to the target task. The results show that learning a superset of required invariances does not harm transfer performance but that missing required invariances degrades transfer performance.

