# OpenReview forum: "Understanding the Role of Invariance in Transfer Learning"
_TMLR — Accepted by TMLR_

### Review · Reviewer_pq9a · 2024-04-08

**Summary Of Contributions:**

This paper designs experiments to study the effect of transformations (such as rotate, hue, blur) on images to transfer learning. The authors claim that invariance to the transformations is important to transfer learning. To investigate the invariance carefully, the authors design a family of synthetic datasets called Transforms-2D which consists of images with a set of foreground objects, a random background, and a set of transformations. With experimental results on the designed dataset, CIFAR-10, and CIFAR-100, the authors claim that invariance is often as or more important than other factors such as the number of training samples, the model architecture, and the class relationship of the training and target task.

**Audience:**

Yes

**Claims And Evidence:**

No

**Requested Changes:**

1. Clarify the main conclusion on what is the role of invariance.

2. Be more specific/rigorous on certain concepts, such as comparing the importance.

3. Some notations need a better explanation (e.g., ones in Section 4.2).

**Strengths And Weaknesses:**

The topic of understanding the role of invariance in transfer learning is quite meaningful and important. The construction of the synthetic dataset is interesting. However, I feel that the main message and the presentation are not good enough.

1. The title is "understanding the role of invariance". However, this paper shows that invariance can either benefit or harm transfer performance (see Section 4.2). This is confusing as it suggests that the role of invariance is uncertain.

2. The target of invariance is vague (i.e., invariance on the source task's input or the target task's input). In the synthetic dataset, transformations are conducted on both the source task's input and the target task's input. The transformation sets can be the same or disjoint. I am not sure what is the specific meaning of invariance in this setup.

3. The conclusion ("invariance is more important than others") is questionable. How do you measure the importance? For example, is there a precise mathematical definition of the importance of the number of training samples that can be quantitatively calculated?

4. Notations in Section 4.2 are confusing. For example, in the second paragraph of Section 4.2, I don't know the meaning of $(X=C,Y=C)$. What is $C$? How can the features $X$ and the labels $Y$ equal to the same thing?

5. For the synthetic dataset Transforms-2D, why do you need a random background? I guess the transformation is only on the foreground object?

---

> ### Author Response · Authors · 2024-04-23
>
> We thank the reviewer for their thoughtful feedback and questions! We address the points below.
> We also uploaded a revision of the paper based on the provided feedback.
>
> ### W1 & C1: Unclear message about the impact of invariance
> The paper argues that invariance significantly affects how well representations transfer to downstream tasks. However, as we demonstrate, the effects of invariance on transfer performance can be both positive and negative and depend on the relationship between the pretraining and target tasks. In Section 4.1 we show that invariance to the right feature transformations can significantly boost transfer performance, while invariance to the wrong transformations can significantly decrease transfer performance. In Section 4.2 We show that invariance to the relevant features themselves can have an even more detrimental effect on transfer performance. Therefore, the contribution of our work in understanding the role of invariance lies in showing its (relative) importance, as well as investigating the conditions under which it benefits and harms transfer performance.
>
> We added a discussion in Section 4 and updated the conclusion section to discuss these takeaways more clearly, in the revised version of the paper.
>
> ### W2: Transformation and invariance setup unclear (in Section 4.1)
> Under the setup in Section 4.1 both the source task and the target task are Transforms-2D datasets with specific sets of transformations acting on the objects. E.g. the target task might transform objects by rotating, blurring and posterizing them. In order to perform well on this task, a model needs to learn representations that are invariant to these transformations, i.e. in the example invariant to rotations, blurring and posterization. Models that are pretrained on source datasets with the same transformations acquire these invariances, whereas models that are trained on datasets with disjoint transformations, e.g. on a dataset where objects are translated, scaled and color inverted, would not acquire the invariances required for the target task. The setup where we pretrain pairs of models, one on a dataset with the same transformations as the target task, and one on a dataset with different transformations, and then transfer both to the target task lets us study the impact of invariance on the performance of the representations.
>
> We added an example in the description of Section 4.1 to clarify the setup in the revision. Please let us know if you think the setup is still unclear.
>
> ### W3 & RC2: Which metric is used to compare the importance of invariance with those of other factors?
> We quantify the importance of invariance as well as other factors, and compare them in Figure 3. Importance here means the difference in performance on the target task due to varying the respective factor. The orange and blue bars show the difference in transfer performance due to varying the non-invariance factors (class relationship, architecture, number of pretraining samples), and the black bars show the impact of being invariant to the same vs different transformations as the target task. By comparing the relative magnitudes of the invariance and non-invariance bars, we can assess the relative performance of the different factors on transfer performance. For the Transforms-2D dataset, the difference due to invariance exceeds that due to other factors in almost all cases. For the CIFAR datasets the impact of invariance is comparable (in some cases a bit lower, and in others a bit higher) in most cases, with the exception of the number of pretraining samples where it tends to be lower. However, as we discuss towards the end of the section, because the CIFAR datasets do not allow us to control all factors of variation, it is possible that the true importance of invariance may be higher than what we measure in our experiments.

---

> ### Author Response · Authors · 2024-04-23
>
> ### W4 & RC3: Unclear notation (X = C, Y = C) in Section 4.2
> In Section 4.2, $C$ stands for CIFAR and $O$ for objects. $X = C$ means that the input data is CIFAR backgrounds, and $Y = C$ means that the task is to classify images based on the CIFAR classes. $X = C + O, Y = C$ means that the inputs are CIFAR backgrounds with objects pasted on them, with the task of classifying the inputs based on their CIFAR classes.
>
> We added a clarification of the meaning of the overloaded $C$ and $O$ notation in Section 4.2 in the revision of the paper.
>
> ### W5: Why does Transforms-2D use random background images?
> Indeed, the random background images in Transforms-2D are not strictly necessary for our analysis. It would also be possible to use the same background image or a monochromatic background for all samples. We nonetheless use randomly sampled backgrounds in order to make the images sampled from Transforms-2D more similar to those in real-world classification datasets, where the classification targets are objects that appear on different backgrounds. The goal is for our findings on Transforms-2D to translate as closely as possible to real-world datasets. The random background images can be seen as another transformation, but since we apply it equally to all samples from Transforms-2D, it can be ignored in the same-vs-different transformation analysis. In other words, the use of random backgrounds helps us make Transforms-2D more realistic, while still retaining precise control over all transformations.

---

### Review · Reviewer_Hw8K · 2024-04-09

**Summary Of Contributions:**

The paper studies the role of invariance in transfer learning. Transfer learning is the problem where one learns a problem structure on one setup, and tries to transfer that knowledge to a different setup. The most common use today of transfer learning comes from the rise of foundation models, where models trained on huge pretraining datasets are then fine-tuned for specific downstream tasks, and often achieve SOTA performance.

On the other hand, good models should be invariant to nuisance or “class-preserving” transformations: a dog slightly shifted in the image to the left still remains a dog and the model should be able to identify it as such.

This paper studies the relationship between these two questions. The key contributions are:
1. Proposing a simple benchmark of synthetic images, Transforms-2D, that precisely controls the objects and transformations in particular images, to be able to study invariance to transformations in a controlled manner.
2. Studying the importance of invariance in transfer learning.
3. Providing an adversarial example where the model being invariant to certain transformations can actually harm its performance in the downstream task.
4. Studying invariance transfer under distribution shift.

**Audience:**

Yes

**Broader Impact Concerns:**

No broader impact concerns

**Claims And Evidence:**

Yes

**Requested Changes:**

In general, I like this paper: I would vote for acceptance if all my questions above are answered to my satisfaction.

**Strengths And Weaknesses:**

**Strengths**

1. The paper is well written and organized.
2. The paper’s key contribution: providing a controlled environment for testing invariance learning in image classification setup, is an interesting contribution in my opinion. The paper’s dataset, if made publicly available, can help the community in studying further aspects of this problem.
3. Thorough experiments and ablations. Understanding the role of learning invariance for transfer learning is an important question, this paper is a good starting point.

**Weaknesses**

**Question 1: Penultimate layer being the most important for transfer learning**

> While the methodology in this paper is applicable to representations at any layer, we focus on the representations at the penultimate layer, since they are most relevant in the context of transfer learning.

Could the authors justify this claim? Prior work [1, 2] has shown that the choice of layer for transfer learning depends on the particular distribution shift between training and test distributions, sometimes earlier layers are better, sometimes middle and sometimes last. This claim needs to be justified or modified in the paper.

**Question 2: Restricting the analysis to “class-preserving” or nuisance transformation**

I think the definition and handling of transformations in the formal description 3.1, while defining invariance, need to be clarified. Invariance is not always a good thing, sometimes transformations are not “class-preserving” or nuisance, and if the model is invariant to these transformations, it can be bad. Imagine the scenario of detecting digits (0 to 9) from images, and the transformation set $T$ be all possible rotations. Then being invariant to $T$ means the model’s representation for 6 and 9 has to be the same, which is obviously bad. Similar things can happen in real life scenarios as well (image two types of snakes which can only be differentiated by their color/habibat (i.e., background) and invariance here can be a bad thing). The analysis needs to be restricted to “class-preserving” transformations and this need to be formally defined in this section.

**Question 3: Set of transformations T form a group**

Does the set of transformations, $T$, form a group?

**Question 4: Transfer learning scenario studied is very simplistic**

The transfer learning scenario tested on this paper is the following:
Train a neural network on source domain
Freeze the model weights, train a linear classifier on the embeddings obtained on the target domain.

This is the linear probing setup described in [3]. It is a very primitive setup, with already known drawbacks. Refer to table 1 and 2 of [3]. Some additional transfer learning setup, such as full fine-tuning (FT), linear probing then full fine-tuning (LP-FT), surgical fine-tuning [1] would make the paper more compelling. How invariant do the representations remain in each setup? How important is invariance as a factor for all of these different transfer learning algorithms?

**Question 5: Plot 2**

Could the authors use 4 different colors instead of 2 colors and dot/dash separation? Current plot is a little bit hard to read.

**Question 6: Dataset diversity**

>  For example, for supervised learning it may be beneficial to spend more effort in obtaining a more diverse set of samples per class, rather than smaller sets of samples for more classes.

I have questions regarding this point.
Do the authors mean unsupervised pre-training or supervised training where the model is trained to learn a function from image x to label y?
Would this model be fine-tuned for a different classification task down the line, or used as is? If the latter, then the number of classes is fixed by the application. If the former, the relative benefit between having more classes present on the pretraining dataset helping the model to learn discriminative features between many different types of objects, vs more diverse samples from the same class is not clear to me. Could the authors point to the exact experimental insight that point to this?

**Question 7: Discussing Prior Work**

Could [4] be discussed by the authors?

**References**

[1] Surgical Fine-Tuning Improves Adaptation to Distribution Shifts, https://arxiv.org/abs/2210.11466

[2] Less is More: Selective Layer Finetuning with SubTuning, https://arxiv.org/abs/2302.06354

[3] Fine-Tuning can Distort Pretrained Features and Underperform Out-of-Distribution, https://arxiv.org/abs/2202.10054

[4] Invariance is Key to Generalization: Examining the Role of Representation in Sim-to-Real Transfer for Visual Navigation, https://arxiv.org/abs/2310.15020

---

> ### Author Response · Authors · 2024-04-23
>
> We thank the reviewer for their very detailed and thoughtful feedback! These are great points which we address below.
> We also uploaded a revision of the paper based on the provided feedback.
>
> ### Q1: Penultimate layer being the most important for transfer learning
> > While the methodology in this paper is applicable to representations at any layer, we focus on the representations at the penultimate layer, since they are most relevant in the context of transfer learning.
>
> This sentence implicitly makes two statements, about 1) the applicability of the methodology in the paper, and 2) the reason for the choice of representations in the analysis.
> 1) The methodology in the paper is used to create invariant representations and measure representational invariance and its downstream effects via controlled data transformations. We can indeed use the same setup to target any layer in the model.
> 2) One of the main goals of the paper is to answer the question “How do the invariance properties of representations affect their utility?”. To investigate it, we fix the feature extractor that produces the representations. If we would update the feature extractor, e.g. via full model fine-tuning, we would change the invariance properties of the representations, and thus would not be able to cleanly answer the question anymore. Since the most common use of fixed feature extraction is at the level of the penultimate layer (e.g. [1, 2, 3]), we also focus on this scenario.
> We realize that this distinction and the reasons for the choice of our setup are not sufficiently clarified, and we updated the above statement in the revision of the paper to be clearer.
>
> ### Q2: Restricting the analysis to “class-preserving” or nuisance transformation
> We agree that invariance can also harm the performance of representations. However, we would like to not explicitly limit the paper to studying class-preserving transformations, since studying non-class-preserving transformations can provide insights into the “failure-modes” of representational invariance. We demonstrate one such related failure-mode in Section 4.2. There, transforming images by adding irrelevant information during pretraining (objects pasted on CIFAR backgrounds, or vice versa) results in representations that are invariant to the added information, effectively making them non-class-preserving when they are used to detect the added information later (e.g. objects). By including such transformations, we can study both the helpful and detrimental effects of representational invariance.
> The transformations used by Transforms-2D are, however, class-preserving, since for all of them there are models that achieve ~100% accuracy (see Section 4.1).
>
> ### Q3: Set of transformations T form a group
> The set of transformations $T$ can, but does not have to, form a group. Some of the transformation types in Transforms-2D form a group, such as rotation and translation, but others don’t such as Gaussian blurring or grayscaling (see Figure 5, Appendix A.1), because they lack inverse transformations. For our analysis it is not necessary that $T$ is a group, e.g. the representations of a model could become invariant to Gaussian blurring while still being useful for downstream tasks, such as when only color information is required, and similarly for grayscaling and shape information.

---

> ### Author Response · Authors · 2024-04-23
>
> ### Q4: Transfer learning scenario studied is very simplistic
> As we already discuss in the response to Q1, point 2), the focus of this paper is on understanding how the invariance properties of fixed representations affect their downstream utility, relative to other factors. We conduct all our analysis with fixed feature extractors, because updating the feature extractor, e.g. via full fine-tuning, linear probing then full fine-tuning [4] or surgical fine-tuning [5], would change the properties of the pretrained representations, and thus we might not be answering the original question anymore.
>
> Nonetheless, we agree with the reviewer that studying additional transfer learning algorithms can help to better contextualize some of our findings, and to understand how difficult it is to change the invariance properties of pretrained representations. Therefore, we expand the analysis in Section 4.1 to include results on full-model fine-tuning. In particular, we run full fine-tuning (FFT) on the whole model after linear probing, i.e. LP-FT [4]. Since this approach already performs well, we do not study additional fine-tuning approaches, such as surgical fine-tuning [5].
>
> We report our results in Appendix C.2. We find that FFT can indeed compensate for a mismatch in representational invariance acquired during pretraining, i.e. FFT can let the models acquire the invariances of the target task. However, the degree to which this is possible depends on the amount of available fine-tuning data. If data is scarce, the amount of performance that FFT can recover is limited, and even if more data is available, the representations pretrained with the right invariances still perform at least as well as the ones pretrained with the wrong invariances.
>
> ### Q5: Different colors in Figure 2
> We updated the plots in Figure 2 in the revised version to use different colors, and also simplified the legends. We did not change the colors too much, however, in order to keep a visual link between curves belonging to the same vs the different transformation relationship. Please let us know if you think that the plots are still confusing.
>
> ### Q6: Dataset diversity
> >  For example, for supervised learning it may be beneficial to spend more effort in obtaining a more diverse set of samples per class, rather than smaller sets of samples for more classes.
>
> This statement is about supervised pretraining, and motivated by the comparison of transfer performance dependent on the pretraining classes vs invariance, in Figures 2a), 7a) and 8a) in Appendix C.1, and the summary of these results in Figure 3a). The results show that the importance of pretraining models to be invariant to the transformations in the target task is en-par with the importance of pretraining them on data with a large set of classes or classes related to the target task. Since, as pointed out by the reviewer, training on more classes likely leads to more discriminative features, our results show that the performance of representations can depend as much on their discriminative power towards relevant information in the input as it does on being non-discriminative, i.e. invariant, towards irrelevant information.
>
> We formulated the sentence above in the context of supervised pretraining, since that is where we make our observations, but it is likely that the same relationship extends to un- or self-supervised settings. We updated the above statement in the revised version of the paper to be more specific.
>
> ### Q7: Discussing Prior Work
> Thank you for pointing out this related work [6] to us. We included it into the related work discussion in the revision.
>
>
> [1] Do Better ImageNet Models Transfer Better?, https://arxiv.org/abs/1805.08974
> [2] Do Adversarially Robust ImageNet Models Transfer Better?, https://arxiv.org/abs/2007.08489
> [3] A Simple Framework for Contrastive Learning of Visual Representations, https://arxiv.org/abs/2002.05709
> [4] Fine-Tuning can Distort Pretrained Features and Underperform Out-of-Distribution, https://arxiv.org/abs/2202.10054
> [5] Surgical Fine-Tuning Improves Adaptation to Distribution Shifts, https://arxiv.org/abs/2210.11466
> [6] Invariance is Key to Generalization: Examining the Role of Representation in Sim-to-Real Transfer for Visual Navigation, https://arxiv.org/abs/2310.15020

---

> > ### Comment · Reviewer_Hw8K · 2024-04-24
> >
> > I thank the authors for their detailed rebuttal. I am happy with their answers as they have sufficiently addressed my questions and concerns, and **I recommend acceptance of this paper to TMLR**.

---

> > > ### Author Response · Authors · 2024-04-24
> > >
> > > We thank the reviewer for engaging with our rebuttal and for recommending acceptance! We are happy that our response is addressing your concerns.

---

### Review · Reviewer_QUU6 · 2024-04-15

**Summary Of Contributions:**

This paper aims to study the role of invariance in the context of transfer learning. In service of attaining this end, the authors introduce a family of synthetic datasets they call "Transforms-2D." By carefully synthesizing these datasets the authors are able to train models to exhibit specific invariances in their representation and to evaluate their performance on transfer tasks that require specific invariances. The authors further investigate the connection between invariance and downstream performance and compare it to other factors studied in the transfer learning literature (for example number of training samples, model architecture, etc.). The findings are that invariance of the target task is often as or more important than these other factors. They also explore the transferability of invariance across tasks and find that in many cases, models can transfer a high degree of learned invariance to out-of-distribution tasks. The paper also validates some of these findings on their synthesized datasets on the real-world datasets CIFAR-10 and CIFAR-100.

**Audience:**

Yes

**Broader Impact Concerns:**

This paper briefly mentions exploiting invariance to harm transfer performance, in Section 4.2. This is a fairly speculative section, but I believe it may be worth adding a high-level discussion about this to a separate broader impact statement (since one is currently not present).

**Claims And Evidence:**

Yes

**Requested Changes:**

The writing in this paper is for the most part quite good. Ideas are clearly presented and figures and tables are well set up to illustrate the main points of the paper. I would like for the presentation of the sens-metric to be made more rigorous to address Weakness 2 above; in particular, I would like the authors to include an explicit discussion (perhaps in the appendix) of how well the sens-metric computations were approximated/how many samples were drawn in making this approximation. Additionally, I have compiled a non-exhaustive list of minor corrections below:

- Abstract: "is as or often more important" -> "is as, or often more, important"
- Introduction, first paragraph: Please use \citep for parenthetical citations. For example, "...on their training dataset Salman et al. (2020)." should clearly be "... on their training dataset (Salman et al., 2020)." The same mistake is made in the following sentence. There are several instances of this mistake in the paper (e.g. first sentence of Section 3.2); please be sure to use parenthetical citations instead of in-text citations, where appropriate.
- Introduction, second paragraph: "In the case of transfer learning on the other hand," -> "In the case of transfer learning, on the other hand,"
- Section 4.2, page 7, final paragraph: "Figure 2 compares we effect of invariance on transfer accuracy" -> "Figure 2 compares the effect of invariance on transfer accuracy"
- Section 4.2, final paragraph: "Practitioners should pay special attention to the transformations present" -> "Practitioners should pay special attention to the variations present"

**Strengths And Weaknesses:**

## Strengths

1. As far as I am aware, this work is the first to attempt a systematic study of the role of invariance in transfer learning. The authors provide a fairly carefully constructed set up using synthetic data to come to the findings described above in the summary.

2. The results with respect to importance of invariance as contrasted with other factors for transfer learning (Figure 2) were interesting. I also thought Table 1 was insightful, showing that exposure to target task features a priori could actually be detrimental for transfer, since the model learns to ignore them.

## Weaknesses

1. The study is fairly limited in that most of the investigations are performed via a highly synthetic construction (i.e. the Transforms-2D dataset). The type of invariance studied is specifically of the empirical approximate nature, i.e., invariance is introduced via data augmentations, not of the exact theoretical nature studied by e.g. Cohen et al. (2021). The synthetic datasets have classes that are derived from augmentations of a single given object, i.e., each specific object defines a class, which is not realistic. However, the authors are upfront about this fact in their "Limitations" discussion and they do reproduce a subset of their results for CIFAR-10 and CIFAR-100 augmentations, which is a reasonable extension of their current synthetic results.

2. Some of the metrics, particularly for measuring invariance in representations, are not rigorously presented. In particular, I am referring to the sens metric in Section 3.4 and the computation of average distance between any two random samples. It should be stated that these are approximations of expectations, and depending on the nature of T and O, one may need to sample many pairs $(x_i, x_i') \sim \mathcal{D}(T,O)$ to obtain a good enough estimate. I do not believe the authors provided commentary on how many samples needed were drawn for their own approximations/sens-metric computations.

## Verdict

This work aims to present a systematic study of the role of invariance in transfer learning. The results make use of carefully constructed synthetic datasets, and are fairly novel, as far as I am aware. Although the conclusions may be somewhat limited by the synthetic nature of the constructions, the authors attempt an extension to CIFAR-10 and CIFAR-100 for a subset of their results, and a number of the results are fairly interesting and insightful. I believe the claims made in the submission are well supported, and the findings of the paper would be interesting to some portion of TMLR's audience, specifically, those working on transfer learning. As such, both criteria for acceptance to TMLR are satisfied; ergo, I recommend acceptance of this paper.

---

> ### Author Response · Authors · 2024-04-23
>
> We thank the reviewer for their feedback and are glad they appreciate our work! We address their points below.
> We uploaded a revision of the paper based on the provided feedback.
>
> ### W1: Results are largely based on synthetic data
> We agree that the use of synthetic data via Transforms-2D for most of the experiments is a limitation that might impact the degree to which our results transfer to real-world settings. Our validation experiments with data augmentations on the CIFAR datasets provide some evidence, however, that we can expect similar patterns to hold under more realistic conditions.
>
> We would also like to point out again that studying invariance under realistic conditions is very challenging, because the information about data transformations that is needed to do so, is not present in most real-world datasets. Therefore, the empirical study of invariance, by its nature, has to rely on synthetic data or datasets sampled under laboratory conditions.
>
> ### W2: sens-metric needs to be presented more rigorously
> We reformulated the sens-metric in terms of expectations in Section 3.4, in the revised version of the paper. We approximate each of the expectations as an average over 10,000 sample pairs. We included that number in Section 3.4 as well.
>
> ###  Minor issues
> Thank you for pointing out the list of corrections to us! We corrected the writing issues and made a pass over all citations to make sure they use the \citep format, except for when the citations appear as part of the text.
>
> ### Broader impact statement
> We added a broader impact statement with a brief discussion about the potential misuse and applications of our findings on the ability to exploit invariance to harm the transfer performance of representations.

---

> > ### Comment · Reviewer_QUU6 · 2024-04-30
> > **Official Comment by Reviewer**
> >
> > Thank you for making the requisite corrections and adding a more formal presentation of the sens-metric. I have noted your comments and maintain an accept rating for the paper.

---

### Decision · Action_Editor_G6ZA · 2024-06-03

**Recommendation:** Accept as is

**Comment:**

The paper looks into the role of invariance in transfer learning. The authors look into penultimate layer and do extensive experiments for a introduced synthetic dataset called Transforms-2D dataset. They added limited experiments on cifar 10 and 100 during rebuttal process. The authors addressed vague parts and gaps in the presentation during the rebuttal process.  The provided answers by the authors were clear and satisfactory.

**Audience:**

The findings of this paper are of interest to TMLR community for people working on transfer learning in vision domain and may spark new thoughts for other data modalities such as speech.

**Claims And Evidence:**

The paper investigates the role of invariance in transfer learning in a systematic case for a synthetic dataset they introduce; Transforms-2D dataset. They also reproduce some aspects for Cifar10 and cifar100 datasets.  They provide extensive ablations and compare the role of invariance to other factors at play in transfer learning. The authors investigate the penultimate layer which has been shown in the community to be the most important layer.  In the revision the sens metric is defined clearly enough.